# Quantification of methane emissions in Hamburg using a network of FTIR spectrometers and an inverse modeling approach

**Andreas Forstmaier[1], Jia Chen[1], Florian Dietrich[1], Juan Bettinelli[1], Hossein Maazallahi[2,5], Carsten Schneider[2,4], Dominik Winkler[1], Xinxu Zhao[1], Taylor Jones[6], Carina van der Veen[2], Norman Wildmann[3], Moritz Makowski[1], Aydin Uzun[1], Friedrich Klappenbach[1], Hugo Denier van der Gon[5], Stefan Schwietzke[7], and Thomas Röckmann[2]**

[1]Environmental Sensing and Modeling, Technical University of Munich (TUM), Munich, Germany
[2]Institute for Marine and Atmospheric research Utrecht (IMAU),
Utrecht University (UU), Utrecht, the Netherlands
[3]Deutsches Zentrum für Luft- und Raumfahrt (DLR), Institut für Physik der Atmosphäre,
Oberpfaffenhofen, Germany
[4]Institut für Umweltphysik, University of Heidelberg, Heidelberg, Germany
[5]Netherlands Organisation for Applied Scientific Research (TNO), Utrecht, the Netherlands
[6]Earth and Environment, Boston University, Boston, USA
[7]Environmental Defense Fund, Berlin, Germany

**Correspondence:** Jia Chen (jia.chen@tum.de), Andreas Forstmaier (andreas.forstmaier@tum.de), and
Florian Dietrich (flo.dietrich@tum.de)

**Abstract.** Methane ($CH_4$) is a potent greenhouse gas, and anthropogenic $CH_4$ emissions contribute significantly to global warming. In this study, the $CH_4$ emissions of the second most populated city in Germany, Hamburg, were quantified with measurements from four solar-viewing Fourier transform infrared (FTIR) spectrometers, mobile in situ measurements, and an inversion framework. For source type attribution, an isotope ratio mass spectrometer was deployed in the city. The urban district hosts an extensive industrial and port area in the south as well as a large conglomerate of residential areas north of the Elbe River. For emission modeling, the TNO GHGco (Netherlands Organisation for Applied Scientific Research greenhouse gas and co-emitted species emission database) inventory was used as a prior for the inversion. In order to improve the inventory, two approaches were followed: (1) the addition of a large natural $CH_4$ source, the Elbe River, which was previously not included in the inventory, and (2) mobile measurements were carried out to update the spatial distribution of emissions in the TNO GHGco gridded inventory and derive two updated versions of the inventory. The addition of the river emissions improved model performance, whereas the correction of the spatial distribution with mobile measurements did not have a significant effect on the total emission estimates for the campaign period. A comparison of the updated inventories with emission estimates from a Gaussian plume model (GPM) showed that the updated versions of the inventory match the GPM emissions estimates well in several cases, revealing the potential of mobile measurements to update the spatial distribution of emission inventories. The mobile measurement survey also revealed a large and, at the time of the study, unknown point source of thermogenic origin with a magnitude of $7.9 \pm 5.3\,\mathrm{kg\,h^{-1}}$ located in a refinery. The isotopic measurements show strong indications that there is a large biogenic $CH_4$ source in Hamburg that produced repeated enhancements of over 1 ppm which correlated with the rising tide of the river estuary. The $CH_4$ emissions (anthropogenic and natural) of the city of Hamburg were quantified as $1600 \pm 920\,\mathrm{kg\,h^{-1}}$, $900 \pm 510\,\mathrm{kg\,h^{-1}}$ of which is of anthropogenic origin. This study reveals that mobile street-level measurements may miss the majority of total methane emissions, potentially due to sources

located within buildings, including stoves and boilers operating on natural gas. Similarly, the $CH_4$ enhancements recorded during the mobile survey from large-area sources, such as the Alster lakes, were too small to generate GPM emission estimates with confidence, but they could nevertheless influence the emission estimates based on total column measurements.

## 1 Introduction

Climate change has a profound impact on living conditions and human societies globally. To a large extent, it is driven by strong anthropogenic greenhouse gas (GHG) emissions. Methane ($CH_4$) is the second most prevalent GHG emitted by human activities (Allen et al., 2013). Over a 20-year horizon, the Intergovernmental Panel on Climate Change (IPCC) estimated the global warming potential (GWP) of $CH_4$ to be 84 times larger than that of carbon dioxide ($CO_2$) (Pachauri et al., 2014). Methane has a relatively short atmospheric lifetime of about $9.1 \pm 0.9$ years (Prather et al., 2012), which makes it an attractive target to diminish the warming rates in the short and medium terms.

In urban areas, there are various types of anthropogenic and natural $CH_4$ sources. Anthropogenic sources comprise fossil-fuel-related emissions, such as fugitive emissions from gas pipelines (Schwietzke et al., 2014; McKain et al., 2015), or road transport and combustion of $CH_4$ (Defratyka et al., 2021) as well as biogenic emissions from sewage systems (Fernandez et al., 2022) and wastewater treatment (Maazallahi et al., 2020). Furthermore, wetlands and bodies of water are common natural $CH_4$ emitters. For instance, in Hamburg, Matousu et al. (2017) showed that the Elbe River releases $CH_4$, and other work has shown that wetlands surrounding the Elbe also produce $CH_4$ (Hummel and Eickers, 2022).

Given the range of possible sources, there are various methodologies used to quantify $CH_4$ emissions from gas pipelines, power plants, refineries, and natural sources. To detect leak indications (LIs) for pipelines, frequently mobile measurements are applied, as shown by Maazallahi et al. (2020), who identified 145 LIs (i.e., $CH_4$ enhancements of more than 10 % above background levels) in Hamburg and 81 LIs in Utrecht while measuring $CH_4$ mole fractions at the street level. Data from such mobile surveys can then be further analyzed to quantify emissions from concentration measurements (Weller et al., 2019). Similarly, Phillips et al. (2013) identified 3356 LIs with concentrations exceeding up to 15 times the global background level by mapping $CH_4$ LIs across all urban roads in the city of Boston. Moreover, they associated the LIs with natural gas after analyzing the isotopic signatures. Weller et al. (2018) evaluated the ability of a mobile survey methodology (von Fischer et al., 2017) to detect natural gas leaks and quantify their emissions. Yacovitch et al. (2015) measured $CH_4$ and ethane ($C_2H_6$) concentrations in a mobile laboratory downwind of natural gas facilities in the Barnett Shale region. To quantify emissions from a natural-gas-based power plant in Munich, Toja-Silva et al. (2017) employed differential column measurements (Chen et al., 2016) and a computational fluid dynamics model. A study by Chen et al. (2020) revealed $CH_4$ emissions at a large folk festival, the Munich Oktoberfest, in 2018 using mobile in situ measurements.

Isolated $CH_4$ sources can be quantified best individually, and this can gradually lead to a better understanding of the mix of sources in a certain area. At the city scale, the mix of sources can, however, become quite complex. Moreover, above-ground-level sources, which cannot be picked up very well using ground-based mobile surveys, can play a role in the mixture of total emissions. Thus, quantifying the emissions of larger areas entails the use of modeling frameworks, which incorporate wind information and mixing between a multitude of individual sources.

To determine natural gas emission rates for the Boston urban area, McKain et al. (2015) and Sargent et al. (2021) incorporated a high-resolution modeling framework with a network of in situ measurements of $CH_4$ and $C_2H_6$. Luther et al. (2022) used a network of portable solar-tracking Fourier transform spectrometers (EM27/SUN) along with a Lagrangian particle dispersion model to calculate emissions from coal mining activity in Poland. The EM27/SUN is an instrument commonly used to measure column-averaged dryair mole fractions of $CH_4$ with high precision. Klappenbach et al. (2015) and Knapp et al. (2021) deployed the portable instrument on ships to measure transects of $CH_4$ concentrations across the Atlantic and the Pacific oceans, respectively, and Hase et al. (2015) set up EM27/SUN spectrometers in Berlin to determine emissions of $CH_4$ and $CO_2$.

In 2019, Dietrich et al. (2021) installed the Munich Urban Carbon Column network (MUCCnet), an urban sensor network that constantly measures greenhouse gases with EM27/SUN instruments in a fully automated and long-term manner. The network consists of four spectrometers around the city and one in the center, such that at least one station will always be upwind and another one downwind. The network of solar-tracking spectrometers measures the total column concentration of $CH_4$ and is, thus, sensitive to both near-ground and aboveground sources.

For this study, we temporarily moved part of the MUCCnet infrastructure to Hamburg and operated four of the spectrometers in locations distributed around the city.

With the third-biggest port in Europe (one of the 20 largest in the world (Hafen Hamburg, 2021), Hamburg contains a large industrial area south of the Elbe River, with oil and

gas refineries, and is one of the largest cities in Europe. According to the TNO GHGco (Netherlands Organisation for Applied Scientific Research greenhouse gas and co-emitted species emission database) inventory, 3 % of total $CH_4$ emissions in Germany occur in Hamburg (Super et al., 2020). Previous studies in Hamburg targeted only specific parts of the city or specific sources alone. Matousu et al. (2017) estimated the emissions from one part of the Elbe River. Furthermore, Maazallahi et al. (2020) explored gas leakages using mobile measurements in the mostly residential area north of the Elbe.

In this study, we aimed at a city-scale quantification of $CH_4$ emissions and, thus, complement the column measurements with mobile $CH_4$ surveys in order to get a better understanding of the spatial distribution of sources. Additionally, source type attribution was carried out to discriminate between plumes of biogenic and thermogenic origin.

A popular method to explore the types of sources is measuring the isotopic composition of plumes. Menoud et al. (2020), Lu et al. (2021), and Dietrich et al. (2023) used the isotopic signature to reveal the source type. For $CH_4$, the isotope ratios between $^{13}C$ and $^{12}C$, and between $^{1}H$ and $^{2}H$ are particularly meaningful ($^{2}H$ is also denoted using D for deuterium). Comparing the observed isotope compositions to references from the literature or previous measurements may then indicate the type of sources.

When quantifying $CH_4$ emissions, usually mobile measurements are utilized or the inversion of column/in situ measurements is applied. In this study, we combined both concepts in order to identify and quantify the sources in a top-down approach. We used a sensor network similar to MUC-Cnet and an emission map with updated distributions based on mobile in situ measurements at the street level. The emission estimate is computed based on the updated map and is compared to the estimate based on the original inventory. For instance, in previous work, Lauvaux et al. (2016) and Jones et al. (2021) compared different prior emission maps (priors) to improve modeling. Lauvaux et al. (2016) compared two emission maps for $CO_2$; however, but both of their maps were taken from literature, whereas our emission maps are updated using mobile measurements that were conducted during the campaign. Additionally, we measured the isotopic composition of $CH_4$ in the city center of Hamburg continuously for the campaign period in order to assign enhancements to biogenic or thermogenic sources. To quantify the uncertainty in the modeled wind field, we deployed a Leosphere WIND-CUBE 200S Doppler wind lidar that retrieves vertical profiles of wind direction and speed (Wildmann et al., 2020; Vasiljević et al., 2016).

## 2 Method

In this work, to measure GHG emissions from a large spatial domain and source mix, as is the case for Hamburg, remote sensing and in situ measurements were combined. The remote sensing setup consists of four FTIR spectrometers, which were deployed around the city, as visible in Fig. 1. An in situ $CH_4$ isotope instrument was co-located with the northern spectrometer, and a wind lidar was also deployed to measure wind direction and speed.

### 2.1 FTIR measurements

Our approach to determine urban emissions is based on the differential column methodology (Chen et al., 2016). The column-integrated dry-air mole fractions of $CO_2$, $CH_4$, and carbon monoxide (CO) are measured with the help of at least two solar-tracking spectrometers that are placed upwind and downwind of an emission source. The concentration gradients between these stations represent the emissions that are generated in between. In Hamburg, the setup consists of four spectrometers to ensure that the differential column condition is met for most wind directions and that a meaningful background can be constrained by the inversion framework. As the wind direction is not constant throughout the measurement period, we placed four spectrometers in different locations around the harbor area where the highest emissions are expected according to the TNO GHGco inventory (Super et al., 2020). The TNO GHGco inventory is an European database that includes spatially resolved emission data for $CO_2$, $CH_4$, CO, nitrogen oxides ($NO_x$), and non-methane volatile organic compounds (NMVOCs). The spatial resolution is $1/60°$ for longitude and $1/120°$ for latitude, which represents an area of approximately $1.1 \, \text{km} \times 0.6 \, \text{km}$ in Hamburg. The emissions are divided into 15 gridded nomenclature for reporting (GNFR) sectors. TNO GHGco is currently the highest-resolution GHG emission inventory that is available for Hamburg. For this study, yearly average emission estimates (as recorded in the inventory) were considered.

Between 27 July and 9 September 2021, our four FTIR spectrometers were measuring in Hamburg. From 30 July to 5 September, the instruments were deployed at the locations shown in Fig. 1. Before and after that, side-by-side measurements of the four spectrometers were carried out on a rooftop at the University of Hamburg to make sure that all instruments were properly calibrated to each other (see Fig. A2).

The EM27/SUN instruments were deployed in custom enclosures that protected the spectrometer from rainfall and adverse weather conditions (Dietrich et al., 2021; Heinle and Chen, 2018). These enclosures automatically open when the sun is visible, so that sunlight enters the spectrometer. When rainfall is detected, the system shuts its cover and the spectrometer is protected against precipitation. The instruments are connected to the internet, which enabled us to operate the four spectrometers remotely during a long campaign.

The enclosures were located to the west, south, and east of the center of Hamburg as well as in the center of the city, as visible in Fig. 1. The three sites outside of the city were selected in order to have little point source influence from lo-

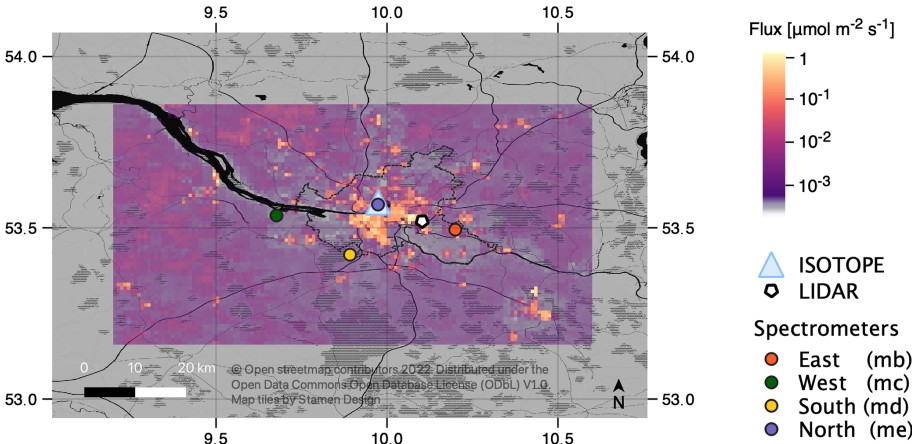

**Figure 1.** Locations of the FTIR spectrometers and the wind lidar during the campaign. The original TNO GHGco emission inventory, which was used as a prior estimate for emissions, is shown for the modeling domain. The border of the administrative region of Hamburg is also shown as a dashed black line. The North (me) spectrometer was co-located with an in situ $CH_4$ isotope instrument. The shaded areas indicate forests and wetlands.

cal, near-by sources, and they were placed about 20 km from each other; therefore, the expected $CH_4$ concentration gradients predicted by the inventory between the stations are well above the instrument precision. The northern site was co-located with the isotope measurements on the rooftop of the University of Hamburg Geomatikum building. This location was chosen by weighting different criteria: firstly, the availability of sites with suitable conditions to house a room-sized setup for isotopic measurements as well as the ability to set up of the FTIR instruments on top of a flat roof; secondly, the requirement for the site to be located outside of an industrial area – a high-emission zone according to the TNO GHGco inventory.

The retrieval of concentrations from interferograms was performed using GFIT GGG2014 (Wunch et al., 2015) according to Dietrich et al. (2021). The measurements of the column-averaged dry-air mole fractions must be properly filtered to exclude measurement errors. In particular, these arise from nonoptimal solar tracking, which is mainly caused by clouds. We used two successive filtering steps. The first filtering step is based on physical properties, such as solar elevation, absolute solar intensity, and solar intensity variation, during a Michelson interferometer scan. The second filtering step uses data statistics to remove outliers and measurement periods with too few data points. In this step, measurements are split when no measurement is available for more than 18 s. Each 2 min section of data is then only considered when continuous measurement data exist for (at least) more than 1 min. This way, outliers from partial cloud coverage during the interferometer scan are reduced. Finally, the remaining continuous measurement sections are averaged using a 10 min moving average filter. Gaps are not filled.

In order to filter out days with fragmented and interrupted measurements due to repeated cloud cover, we only consider

measurement days when at least two stations were measuring at the same time for more than 5 h. In August 2021, the weather was unexpectedly cloudy, and the systems were idle on many days; however, we still had 9 good measurement days with sufficient sunshine to carry out the measurements.

## 2.2   In situ measurements

To support the modeling and the calculation of the final emission estimate, in situ measurements were performed with a Picarro GasScouter G4302, which measures $CH_4$ and $C_2H_6$, and a Picarro G2301 greenhouse gas analyzer, which measures $CH_4$ and $CO_2$. Both sensors were mounted inside a car, and a tube was used to pipe the air from the inlet located on the front bumper into the sensors. The height of the inlet was ca. 60 cm above ground level. The $CH_4$ concentration measurements, which were carried out with a sampling frequency of 1 and 0.3 Hz for the Picarro GasScouter G4302 and Picarro G2301 instruments, respectively, were temporally averaged using a moving average with a 10 s time window. The averaging improves the precision of the $CH_4$ measurements from 3 ppb at a 1 s integration time to 1 ppb at a 10 s integration time (Chen et al., 2020).

In order to verify and update the prior estimate of an emission map derived from the TNO GHGco inventory (Super et al., 2020), mobile surveys were conducted in the city and in the industrial area. The first part of the surveys focused on the residential areas of Hamburg, mostly to the north of the Elbe, and were conducted in the year 2018 by Maazallahi et al. (2020). In 2021, these existing measurements were complemented by a mobile survey with the same instruments in the industrial harbor area (all tracks can be seen in Fig. 2). The new survey took place between 9 and 21 August 2021. During the surveys, $CH_4$, $C_2H_6$, and $CO_2$ concentrations were recorded and mapped with a GPS logger. In order to

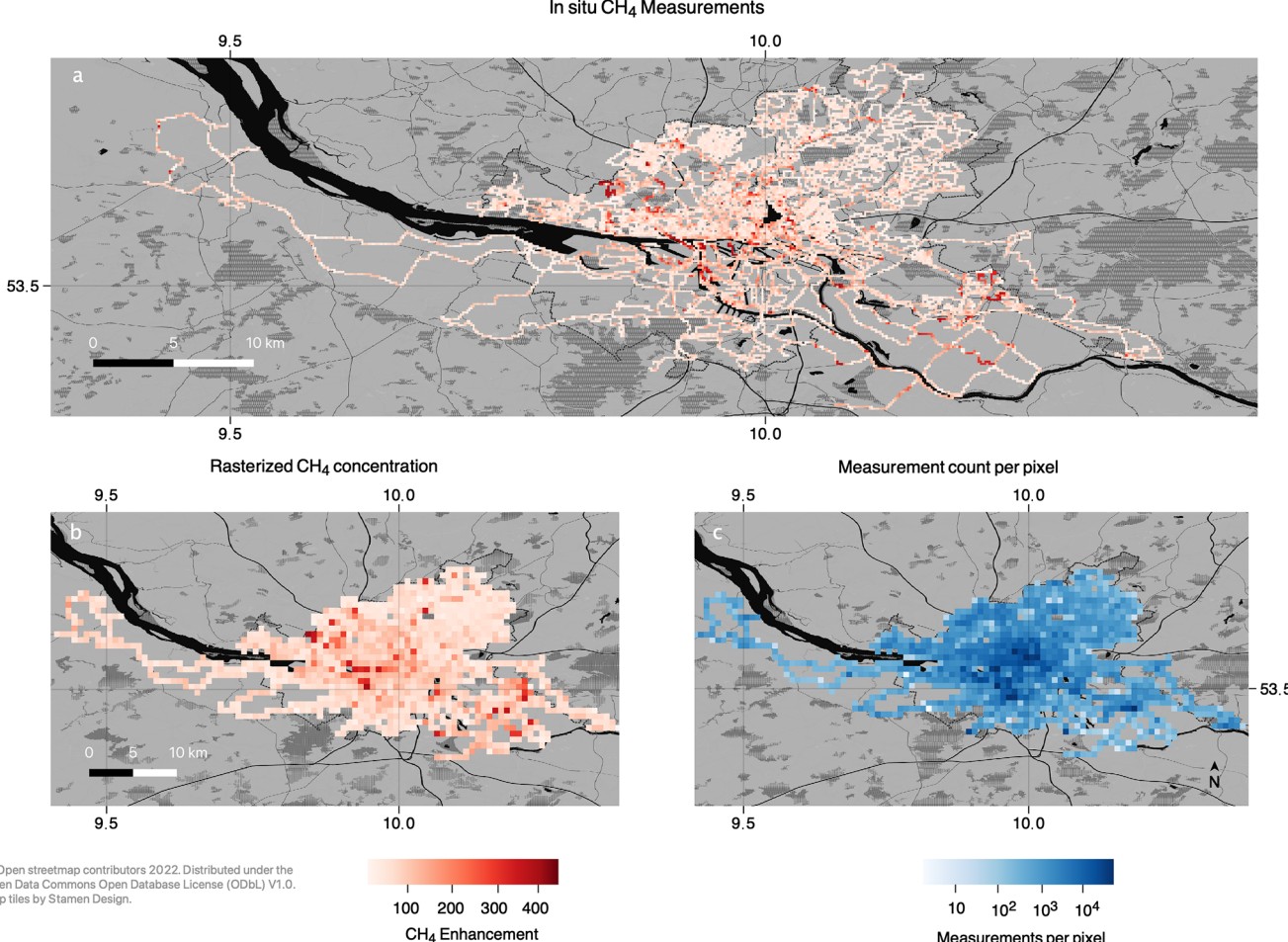

**Figure 2.** The measured $CH_4$ concentrations along the driven tracks recorded during the mobile surveys in 2018 (Maazallahi et al., 2020) and during this campaign in 2021 are shown in the top map. The map in the lower left shows the concentrations rasterized onto the modeling grid. The density of measurement points per modeling pixel is plotted in the lower right panel.

cover the areas in the harbor that were not accessible by public road, a boat was equipped with the Picarro GasScouter G4302 and additional surveys were carried out on the Elbe River and the waterways in the harbor area on 20 August. Some private roads in the harbor areas were sampled after permission was granted from the facility owners, including a wastewater treatment plant and two refineries.

The recorded $CH_4$ concentration during the mobile surveys was separated into its two components: the background and the enhancement peaks occurring near localized sources. While the background is generally rather smooth and varies only slowly with location, the short-time component (peaks in the signal) is caused by emissions from nearby sources. The background signal was determined as the lowest fifth percentile of a $\pm 2.5$ min time window around each data point. In order to compile an improved estimate of the spatial distribution of the emissions, both the complete signal (background and enhancement peaks, later referenced as "upd:all") and the peaks only (later referenced as "upd:elv")

were averaged on the inventory grid, as can be seen in the right and central plots of Fig. 3.

The spatial distribution of emissions recorded in the original TNO GHGco inventory was then updated using the mobile concentration measurements. We assumed that it was more likely that we would find emission sources in regions where we measured high concentrations than in regions where we measured only background concentrations. The emissions of all inventory pixels that were covered by our mobile survey were summed up and distributed according to the measured concentrations, weighted by the number of measurements per pixel.

The following equations show how the original inventory $E(x, y)$, a function of latitude $x$ and longitude $y$, was updated using the concentration measurements $C(x, y)$ averaged on the inventory grid. These measurements were either the whole signal (upd:all) or the peaks only (upd:elv).

First, the concentrations were normalized in the area where measurements were available, and the emissions

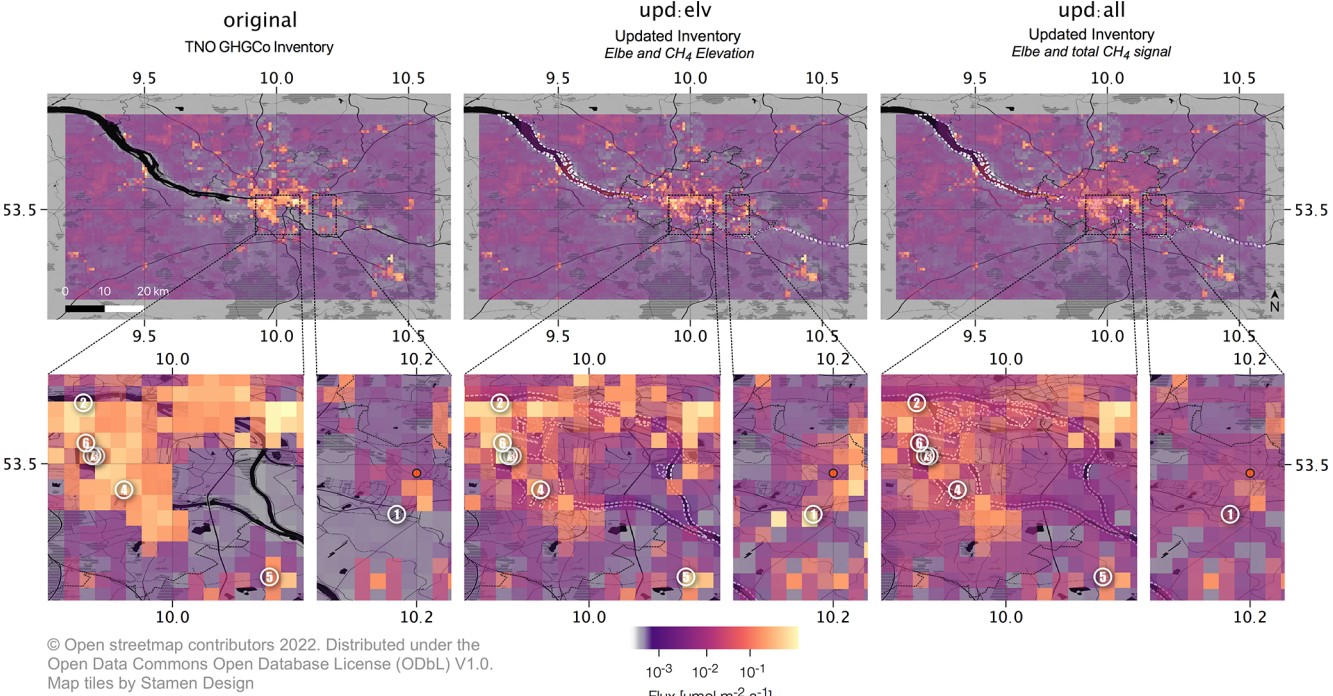

**Figure 3.** For this study, three different inventories were used: (1) the original TNO GHGco inventory, (2) the updated inventory using the measured $CH_4$ enhancement of the mobile survey (upd:elv), and (3) the inventory updated using the complete (background and enhancement) $CH_4$ signal (upd:all). All versions of the inventory include an a priori estimate of the Elbe River derived from the findings of Matousu et al. (2017). In this figure, the TNO GHGco inventory is shown without the Elbe for a better comparison. The close up sections show the locations where point sources were quantified using mobile measurements. Sources 3 and 7 are co-located.

recorded in the inventory were redistributed according to the measured concentrations. The new inventory values $N(x, y)$ were calculated as follows: TS1

$$N(x, y) = \frac{C(x, y)}{\sum_{x,y \in C(x,y) \neq NaN} C(x, y)} \sum_{x,y \in C(x,y) \neq NaN} E(x, y). \quad (1)$$

TS2 A weighting mask $W(x, y)$ was then defined according to the number of measurements per pixel: CE1

$$W(x, y) = \frac{\text{count}(C(x, y))}{\max(\text{count}(C(x, y)))}. \quad (2)$$

New values were mixed between the original inventory value $E(x, y)$ and the new values suggested by Eq. (1) according to the weighting mask $W(x, y)$. In pixels with few measurement points, the new emission value of the pixel was chosen closer to the original value of the inventory. In pixels with many measurement points, the value was chosen closer to the value suggested by the concentration-based redistribution of emissions.

$$N_{\text{mixed}}(x, y) = E(x, y)(1 - W(x, y)) + W(x, y)N(x, y) \quad (3)$$

The updated inventory $E_{\text{updated}}$ is then calculated depending on the availability of concentration measurements, as follows:

$$E_{\text{updated}}(x, y) =$$
$$\begin{cases} E(x, y), & x, y \in C(x, y) = NaN \\ \\ \dfrac{N_{\text{mixed}}(x, y)}{\sum_{x,y \in C(x,y) \neq NaN} N_{\text{mixed}}(x, y)} \displaystyle\sum_{x,y \in C(x,y) \neq NaN} E(x, y), & x, y \in C(x, y) \neq NaN. \end{cases} \quad (4)$$

For a better comparability between the updated and the original inventory, the sum of emissions in the area covered by our mobile measurements is equal in the original and the updated versions.

The original TNO GHGco inventory has been created using proxy data. For example, all industry emissions reported by Germany were distributed on a map according to the distribution of industrial areas in Germany. In the three inventories used in our study, the industrial area south of the Elbe River has lower emissions in the updated versions than in the original inventory, as these emissions were distributed over a wider area according to the mobile measurements.

Furthermore, an inventory layer containing the Elbe and its estimated emissions was added. Matousu et al. (2017) estimated the $CH_4$ flux from the Elbe into the air for different sections of the river to be between 0.25 and 4.5 $kg\,h^{-1}\,km^{-2}$. The emission values for the Elbe in each grid cell are the

average flux of the corresponding section multiplied by the proportion of the Elbe inside the grid cell. For parts of the Elbe that were not covered in the study by Matousu et al. (2017), the average emissions of the study (2.5 kg h$^{-1}$ km$^{-2}$) were used as a prior estimate.

During the mobile survey carried out to map concentrations of CH$_4$, multiple transects were undertaken through observed plumes. Plumes were manually selected when an enhancement higher than 100 ppb was observed. For each plume and location, between 3 and 15 transects were carried out. Sections in which the measurement car was immobile were removed before further analysis. Emission estimates were derived based on a Gaussian plume dispersion model (GPM), as described in Maazallahi et al. (2020). As the exact emission location was unknown for several sources, we calculated an estimate of the emission location for each transect. For this purpose, the possible Pasquill–Gifford stability classes were first estimated using wind sensor data, information on the planetary boundary layer height, and information on the surrounding area. In a next step, the width of the plume was used to derive $\sigma_y$. Under the assumption of a constant stability class, $\sigma_y$ can be expressed as a function of the distance to the source. This function is independent of the flux, and thus an estimate for distance can be determined. The distance in combination with the wind direction lead to an estimate of the location for each transect. For each estimated source location, the flux is then estimated. Errors in the location estimates are propagated into the mean emission estimate. For each location, all relevant Pasquill stability classes were estimated. The presented mean emission estimates are the average of estimates obtained for each relevant stability class and location estimate.

## 2.3 Wind measurements

During the transect drives that were carried out for emission quantification, wind information close to the ground was important. Therefore, a local portable wind sensor (Lufft WS200-UMB smart weather sensor), which measured wind direction and wind speed at an altitude of 2 m, was deployed.

To evaluate the uncertainty in the atmospheric transport of the ERA5 model inside the modeling domain, a Leosphere WINDCUBE 200S Doppler wind lidar was deployed at the weather mast in Billwerder, Hamburg (see Fig. 1). The lidar provides a wind profile from approximately 80m to the top of the atmospheric boundary layer. Measured wind direction and wind speed were compared to the ERA5 model data for all altitudes where model and lidar data were available (see Fig. A8). For each measurement day, the standard deviation of the differences between the ERA5 model and the lidar wind direction and speed were derived.

## 2.4 Isotope measurements

We took continuous measurements of CH$_4$ at an inlet height of 80 m on the rooftop of the University of Hamburg Geomatikum building. Measurements started on 2 August and the setup was operational for most of the campaign period. It was shut down once for maintenance on 25 August and resumed operation on 27 August. We deployed an isotope ratio mass spectrometer (IRMS) that continuously measured $\delta^{13}$C and $\delta$D with a Delta V Plus and Delta$^{plus}$ XL from Thermo Fisher Scientific (Brass and Röckmann, 2010; Röckmann et al., 2016).

In addition to the continuous measurements, air samples were taken at several locations while carrying out the mobile survey in order to characterize the source types of observed plumes, similar to Menoud et al. (2021).

To investigate the source mix of the measured CH$_4$ and to decide whether it was mainly of thermogenic or biogenic origin, continuous analysis of the dual stable isotopic composition of CH$_4$ ($\delta^{13}$C and $\delta$D) was performed, similar to previous studies (Röckmann et al., 2016; Menoud et al., 2020, 2021). $\delta^{13}$C values are reported vs. the Vienna Pee Dee Belemnite (VPDB) standard, and $\delta$D values are reported vs. the Vienna Standard Mean Ocean Water (VSMOW) standard.

The dominant source type that is responsible for the observed CH$_4$ elevations above background in Hamburg was obtained by comparing $\delta$D and $\delta^{13}$C values obtained from a Keeling plot analysis (Keeling, 1958) to similar sources signatures in the literature.

Sources were classified as biogenic when $\delta^{13}$C values were between $-45\,‰$ and $-90\,‰$ and $\delta$D values ranged from $-245\,‰$ to $-360\,‰$. In contrast, signatures with $\delta^{13}$C values between $-32\,‰$ and $-67\,‰$ and a $\delta$D value between $-118\,‰$ and $-200\,‰$ were attributed to thermogenic emissions (Röckmann et al., 2016).

## 2.5 Inverse modeling approach

In order to quantify the urban emissions based on the concentration measurements, a Bayesian inversion framework was used. We utilized and adapted the model as presented in Jones et al. (2021) according to the specific requirements of the Hamburg urban area.

This model was designed to quantify diffuse emission sources with the help of several ground-based spectrometers, such as the EM27/SUN. The model accounts for temporal variations in the background concentrations using the so-called background influence matrix (BIM). Analogous to Jones et al. (2021, their Supplement S1), virtual particles are released along the line of sight according to the given solar azimuth and elevation angle at 13 altitudes up to 2220 m height above the instrument. These particles are released at the receptor time and travel backwards in time until they reach the simulation border (background time). A weight-

ing factor is assigned to the times when the particles cross the border (background time), based on the number of particles passing the border at that time. This results in a nearly Gaussian-shaped distribution of background time for each receptor time. Every 15 min, such a release of particles from each receptor station is initiated. Releasing particles backwards in time is also the basis to generate footprint matrices, which represent the influence of all locations in the domain on the measurement site at a certain receptor time. The footprint is the summation of the residence time of all of the particles in a grid cell.

In order to generate those backward trajectories and the footprints, the STILT (Stochastic Time-Inverted Lagrangian Transport) model is used. The meteorological input data for this model were provided by the ERA5 data set (Muñoz Sabater, 2019).

The TNO GHGco inventory was used as a prior emission map (Super et al., 2020). Additionally, the updated inventories that are depicted in Fig. 3 and described in Sect. 2.2 were compared.

Further assumptions for the model are a spatially homogeneous concentration at the modeling boundary (concentrations can vary with time) and a known spatial distribution of the diffuse emission sources provided by the inventory. The model minimizes a cost function to find the scaling factor for each emission sector that best fits the model to the measurements.

The cost function is described as follows:

$$\mathcal{J}(\boldsymbol{x}) = \frac{1}{2}(\mathbf{K}\boldsymbol{x} - \boldsymbol{y})^T \mathbf{S}_\varepsilon^{-1}(\mathbf{K}\boldsymbol{x} - \boldsymbol{y})$$
$$+ \frac{1}{2}(\boldsymbol{x}_a - \boldsymbol{x})^T \mathbf{S}_a^{-1}(\boldsymbol{x}_a - \boldsymbol{x}), \tag{5}$$

where $\boldsymbol{x}$ is the unknown that needs to be fitted and that contains the information of the scaling factors for different emission sectors and the background concentration, $\mathbf{K}$ is the sensing matrix that contains the footprints' information and BIM, $\boldsymbol{y}$ is the column concentration measurements obtained from the four EM27/SUN instruments, $\boldsymbol{x}_a$ is the prior emission information, and $\mathbf{S}_a$ and $\mathbf{S}_\varepsilon$ are the prior error covariance matrices for the prior emission and data–model mismatch, respectively.

In this study, we use the existing framework developed by Jones et al. (2021) to estimate the emissions for individual days. The total emission estimate for the campaign period was calculated as the weighted average of the individual day results. The average was weighted by the number of measurement points per day. Negative emissions were considered when forming the average.

Emission estimates for smaller areas, such as the city of Hamburg or the northern part of Hamburg, were calculated by summing up the prior emissions from inventory pixels in that region. This sum was then multiplied by the inversion result (scaling factor) for all days from the respective inventory.

## 2.6 Uncertainty assessment for the inverse model

The error assessment follows the approach described in Jones et al. (2021). The uncertainties are extracted from the posterior covariance matrix $\mathbf{S}_p$, which is mathematically computed based on the sensing matrix $\mathbf{K}$ and the prior error covariance matrices $\mathbf{S}_\varepsilon$ and $\mathbf{S}_a$:

$$\mathbf{S}_p = \left(\mathbf{K}^T \mathbf{S}_\varepsilon^{-1} \mathbf{K} + \mathbf{S}_a^{-1}\right)^{-1}. \tag{6}$$

The uncertainty in the observations $\left(\sigma_{\text{observation}}^{\text{prior}}\right)$ was chosen as the sum of the instrument precision, which is 0.2 ppb when the measurements are integrated over 10 min (Chen et al., 2016), and the transport error calculated for each day. The transport error was obtained by simulating a set of footprints for different wind directions. The wind directions were drawn from a normal distribution, with a standard deviation derived for each day by comparing the wind direction of the lidar and ERA5 model. No variations were made for the wind speed, as the mean mismatch between the lidar and model was as low as $0.49 \, \text{m s}^{-1}$. The resulting set of footprints was then multiplied by the three inventories used in this study to obtain a distribution of prior expected enhancements for all possible wind directions. The standard deviation of this distribution was used as the transport error. $\sigma_{\text{observation}}^{\text{prior}}$ values are the diagonal elements of $\mathbf{S}_\varepsilon$.

The uncertainty in the prior emission map $\left(\sigma_{\text{sector}}^{\text{prior}}\right)$ was chosen separately for the river layer and the layer with all anthropogenic sources. The river was given an uncertainty of $\pm 200\%$, whereas the anthropogenic sector was given an uncertainty of $\pm 100\%$. The uncertainty was higher for the river because a priori information was only available for a section of the river in Matousu et al. (2017) and other areas were estimated with a mean flux of $2.5 \, \text{kg h}^{-1} \, \text{km}^{-2}$. The uncertainty in the background $\left(\sigma_{\text{background}}^{\text{prior}}\right)$ was chosen to be 8 ppb, slightly below the value of 10 ppb used by Jones et al. (2021), according to a comparison of MUCCnet measurements with the Copernicus Atmosphere Monitoring Service (CAMS) data. $\sigma_{\text{sector}}^{\text{prior}}$ and $\sigma_{\text{background}}^{\text{prior}}$ are the diagonal elements of $\mathbf{S}_a$, as in Jones et al. (2021).

## 3 Results

### 3.1 Wind measurements

The model mismatch for wind direction and wind speed was calculated for the selected measurement days by comparing lidar data and ERA5 model data. Table 1 shows that the wind speed is generally matched well by the model. A mean difference of $0.49 \, \text{m s}^{-1}$ between the model and lidar was recorded (i.e., the lidar recorded a slightly faster wind speed on average). The wind direction is off by an average of $6.0°$.

The calculated mismatch was considered when calculating the transport error for each day, as recorded in Table A2.

**Table 1.** Comparison of ERA5 and lidar wind data.

| | Wind speed model mismatch (m s$^{-1}$) | | Wind direction model mismatch (° CW) | |
| --- | --- | --- | --- | --- |
| Date | Mean | SD | Mean | SD |
| 6 August 2021 | 1.1 | 1.1 | −2.5 | 24 |
| 11 August 2021 | −0.06 | 0.91 | 12 | 20 |
| 12 August 2021 | −0.07 | 0.58 | −4.8 | 20 |
| 23 August 2021 | 0.70 | 0.66 | 6.0 | 6.5 |
| 24 August 2021 | 0.13 | 0.53 | 13 | 11 |
| 31 August 2021 | 0.05 | 0.70 | 15 | 10 |
| 1 September 2021 | 1.1 | 0.52 | −2.5 | 13 |
| 3 September 2021 | 1.2 | 0.51 | 6.3 | 8.1 |
| 5 September 2021 | 0.30 | 0.57 | 13 | 13 |
| Mean | 0.49 | 0.7 | 6.0 | 16 |

To compute the standard deviation and mean of the mismatch of wind direction and wind speed between the ERA5 model and lidar data on the selected measurement days, the model values have been subtracted from the lidar values. CW stands for clockwise. CE2

These daily transport error values were then considered in the inversion framework.

During the campaign period, there was a good agreement between the modeled and measured planetary boundary height, as can be seen in Fig. A3.

A comparison of wind data from the local sensor (at 2 m altitude) used for the GPM emission estimates and the weather mast (at 10 m altitude) showed a mean difference of 1.6 m s$^{-1}$ (standard deviation of the difference was 1.2 m s$^{-1}$) for the wind speed and a mean difference of 15° CW (standard deviation of 31° CW) for the wind direction.

## 3.2 Column measurements

In Fig. 4, the measured concentrations as well as the modeled signal and background are shown for each day. The corresponding emission estimates for the original inventory and the two updated inventory versions are shown in Fig. 5.

On 2 respective days CE3, 23 August and 3 September, while the stations were measuring simultaneously, little enhancement ($< 2$ ppb) between the stations was observed for most of the time. This is visible from the measurements of the different spectrometers (plotted as colored dots) in Fig. 4. Small enhancements result in low emission estimates for these days, as can be seen by comparing Figs. 4 and 5.

On other days, in general, larger enhancements between the stations were observed, resulting in larger emission estimates. The 6, 11, 12, and 24 August all show emission estimates higher than or equal to the prior. The 1 and 5 September have been estimated at values between the prior and zero emissions.

Looking at the result for 12 August in Fig. 4, it becomes evident that the North (me) spectrometer measured a peak at around 12:00–13:00 UTC; this peak was not measured by the other stations. During the time of the peak, the wind did not change direction and was constantly blowing from the south. Thus, the prominent elevation indicates the presence of an unknown temporary source. The inversion framework assigns this elevation to an enhancement of the background concentration (dashed black line at around 10:15 UTC) to balance out observations and prior expected contributions.

On 31 August, the West (mc) spectrometer, which was located about 1.5 km south of the Elbe River, measured an enhancement of about 5 ppb compared with the other stations throughout the whole day, as can be seen in Fig. 6a. During the course of this day, the wind came from the north, as can be seen by looking at the footprints visualized in white and blue on top of the TNO GHGco inventory in Fig. 6b.

With the original inventory, the inversion cannot model the enhancement seen by the West (mc) station, as there is no large source in the inventory north of the spectrometer. In such a case, the modeled signal, visualized using solid lines in Fig. 6a, does not match the actual measurements (dots) very well. The difference is visible, for instance, by looking at the distance between the purple line (the West (mc) station) and the purple dots in Fig. 6. In such a case, the modeled background at the domain boundary (dashed black line) is fitted higher than the signal (solid lines). This can result in negative emission numbers (Fig. 6c), as the enhancement (measurements − background) becomes negative for most time steps.

When the Elbe River is included in the emission inventory as quantified by Matousu et al. (2017), the modeled signal fits the measurements better and the inversion result returns positive emissions. On this particular day, the emissions of the Elbe were quantified as being much higher than the a priori annual emissions of the Elbe in our domain (350 kg h$^{-1}$). Thus, for consistency, we decided to include the Elbe River in all other model runs and days presented in this paper. On other days, the emissions from the Elbe were close to the prior estimate or around zero, as can be seen in Fig. A6.

The expected contributions from different sectors for the 31 August are shown in Fig. 7. These expected contributions were calculated using the footprint and the inventory. As can be seen, the West (mc) station was sensitive to river emissions on 31 August. Moreover, the West (mc) station was sensitive to river emissions on 23 and 24 August as well as on 5 September. On all of these days, the concentrations measured by the West (mc) station were generally higher than those measured by other stations (see Fig. 4).

## 3.3 Correlation assessment

For the selected days, the correlation between modeled and measured CH$_4$ concentrations was very high for the total signal (modeled background + enhancement), as can be seen in

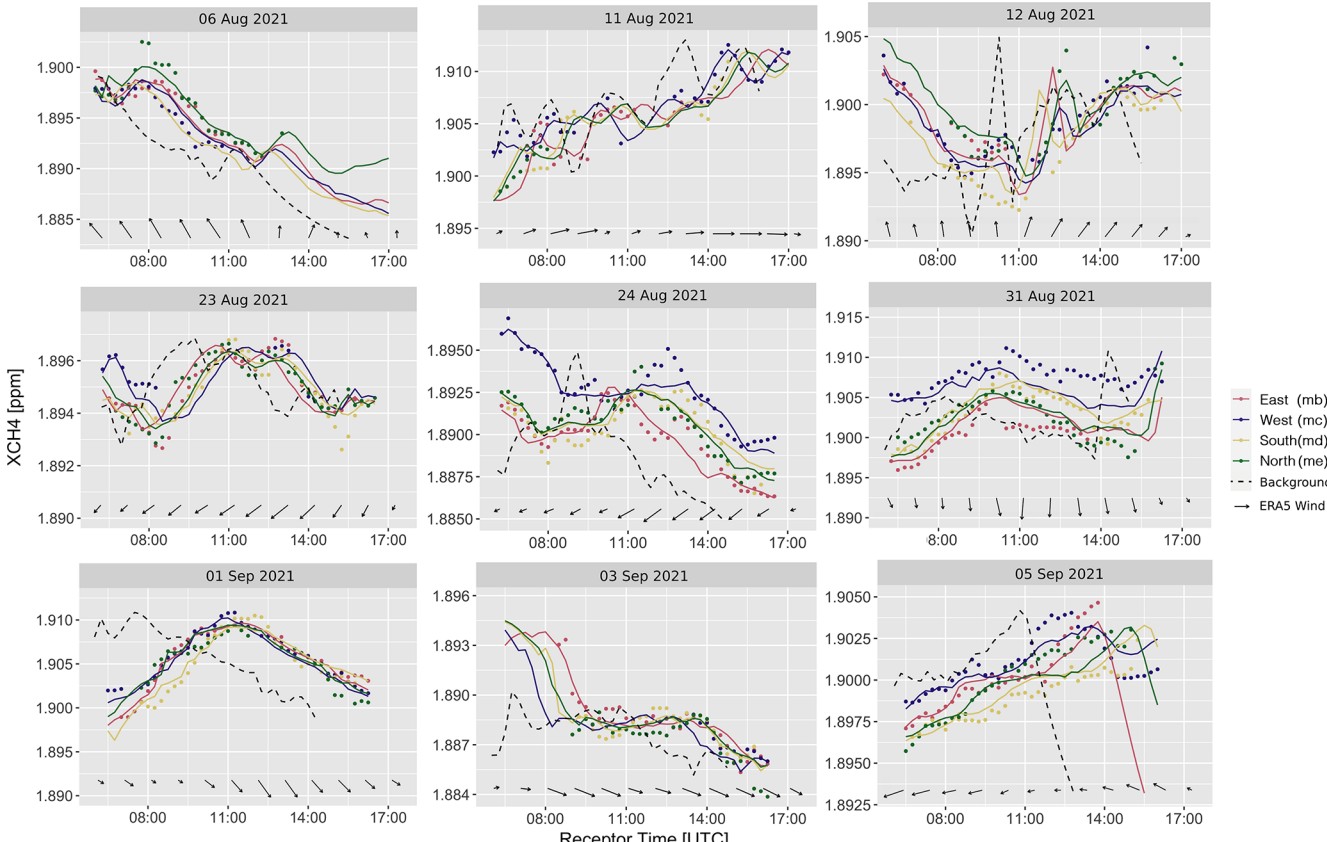

**Figure 4.** Plot of all selected measurement days used in the inversion framework. The measurements are plotted using different colored dots for each station. The colored lines represent the posterior observations generated by the inversion framework, and the dashed black line shows the fitted background at the domain boundary. Wind direction and relative speed are shown as arrows (downward-pointing arrows indicate northerly wind).

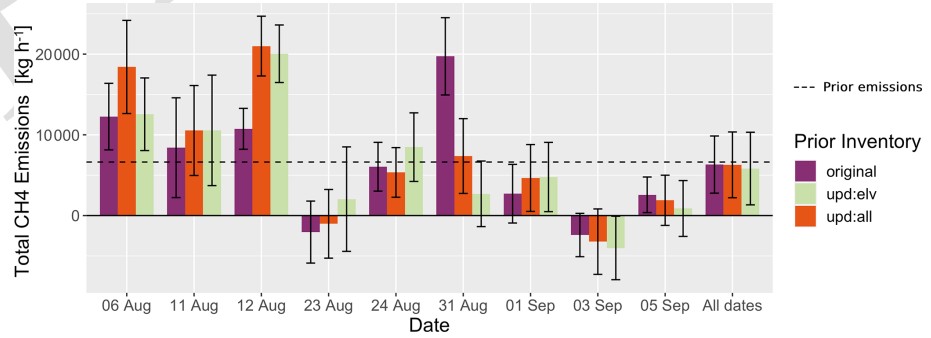

**Figure 5.** Inversion result for all selected days and the three different prior emission inventories: "original" (unaltered TNO GHGco) and "upd:elv" and "upd:all" (updated using mobile measurements and filtered for only the peaks and for the complete measurement signal, respectively). The dashed line represents the prior emission estimate of the TNO GHGco inventory for the modeling domain. The emission of the Elbe River was added to all versions of the emission inventory. The reader is referred to Fig. A6 in the Appendix for the split of total emissions into natural and anthropogenic sources.

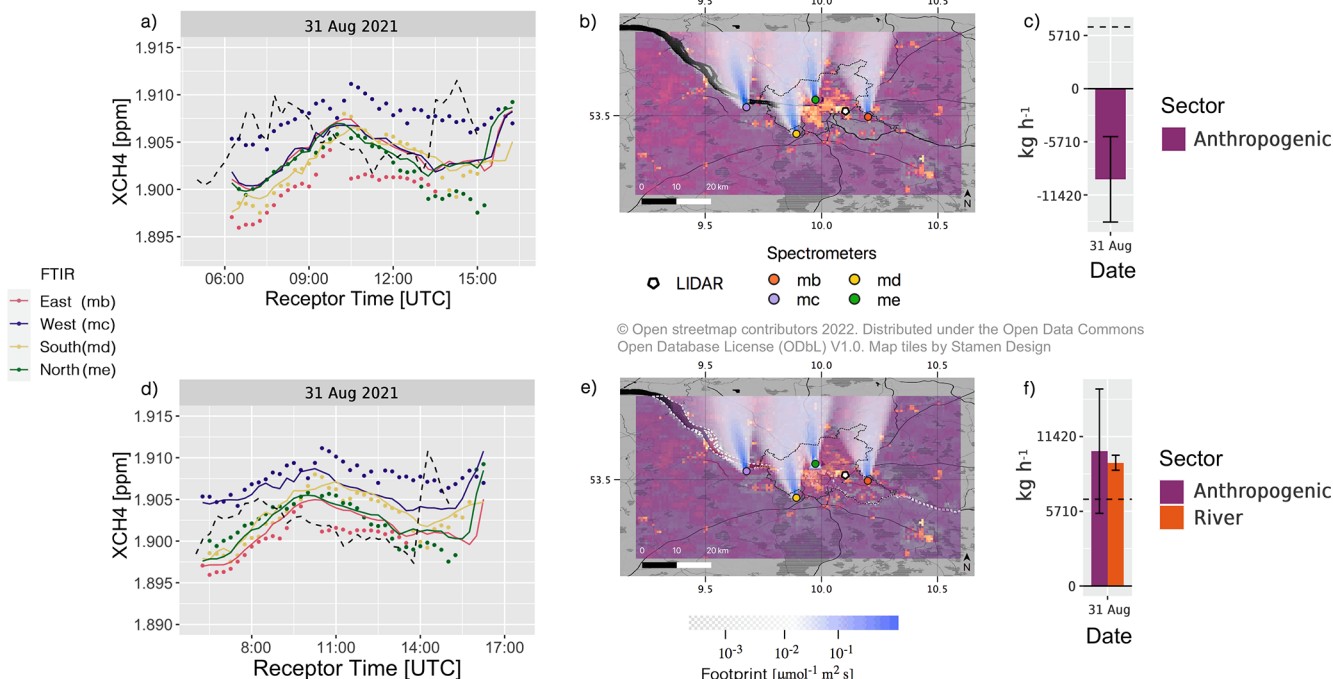

**Figure 6.** Comparison of the inversion result for different priors on 31 August. In panels (**a**)–(**c**), the inversion result for the original inventory is shown. Panels (**d**) to (**f**) show that the inversion result changes from negative emissions (**c**) to positive emissions (**f**) when the Elbe River is added into the emission inventory (compare the river region, outlined with a dashed white line, in panels **b** and **e**). When the river is included in the inventory, the modeled signal (solid line) in panel (**d**) is closer to the measurements (dots) for the West (mc) station than in panel (**a**). The dashed black line shows the fitted background at the domain boundary.

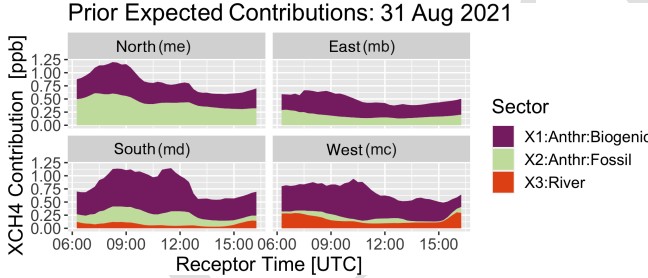

**Figure 7.** Expected prior contributions from different sectors on 31 August for the different stations: North (me) is the northern station, East (mb) is the eastern station, South (md) is the southern station, and West (mc) is the western station.

Fig. 8a. Figure 8b shows the correlation for the enhancement only. Modeled enhancements were divided into 0.5 ppb bins, and sample means of the first bin (0–0.5 ppb) and the all other bins are significantly distinct ($p = 0.001$), demonstrating the quantification of small and large enhancements in total column $CH_4$ (Jones et al., 2021).

The correlation increased significantly when including the natural source into the modeling, as is visible in Fig. A4 in the Appendix.

## 3.4 Comparison of different inventories

In this study, three different versions of the emission inventory have been used as a prior estimate for the spatial distribution of $CH_4$ emissions in Hamburg. While all three versions lead to comparable results for all measurement days combined, the results differ significantly on single days, as can be seen in Fig. 5. Over the course of all 9 d CE4, the footprint has covered almost all of the areas of the modeling domain (because of different wind directions throughout the campaign), whereas only small parts of the domain are covered on single days.

The difference in emission estimates for the three inventory versions on single days can be explained by the different spatial distributions of prior emissions. In the area covered by the footprint on a particular day, the recorded emissions can be different in the original and the updated inventories. These differences in prior emissions for each inventory version lead to different scaling factors with the same observations. The scaling factor is determined by the inversion when scaling the three inventory versions to match the forward model and the observations. As all emission inventories are normalized and have the same total emissions, a different scaling factor applied to the whole inventory can then lead to different total emission estimates.

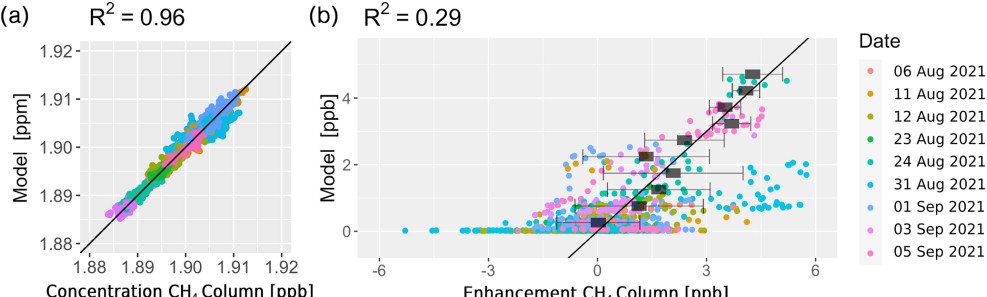

**Figure 8.** Regression plot of the measured and modeled $CH_4$ signal for all 9 selected measurement days. Panel **(a)** refers to the whole signal (background and enhancement), whereas panel **(b)** shows the correlation for the enhancement only. The 1 : 1 line is shown in black. The black rectangles represent the mean values of the modeled enhancement divided into 0.5 ppb bins. The horizontal error bars represent the sample standard deviation in each bin. The mean of the 0–0.5 ppb bin is significantly different ($p = 0.001$) from the mean of all other bins, which shows that small and large enhancements in total column $CH_4$ can be detected and quantified. TS3

For instance, the inversion result can differ between the original and the modified inventories when there is footprint covering the industrial zone of the inventory. This zone has higher emissions in the original inventory than in the two updated versions, as visible in the lower left panel of each inventory in Fig. 3. With the same observations, the scaling factor calculated by the inversion framework will be slightly lower with the original inventory, as the inventory already has higher emissions recorded here, and higher with the updated inventories, as the updated inventories have lower emissions recorded here. For all days, the area covered by the measurement footprints encompasses the domain more uniformly; thus, a result in a similar magnitude can be expected for all inventory versions.

## 3.5 Emission rate estimates from column measurements and comparison to the car-based study

We ran the inversion for all inventories (original, upd:all, and upd:elv) with the Elbe River as a separate sector. Therefore, the emissions are split between river emissions (natural) and anthropogenic sources. We determined the emissions for the entire modeling domain as well as for the area inside the municipality border of Hamburg. The extent of the modeling domain and the area considered to be the city can be seen in Fig. 1.

The emission rate estimates for natural and anthropogenic sources combined in our modeling domain sum to $6300 \pm 3500\,\mathrm{kg\,h^{-1}}$ for the original inventory, $6300 \pm 4100\,\mathrm{kg\,h^{-1}}$ for the updated inventory using peaks and background (upd:all), and $5800 \pm 4500\,\mathrm{kg\,h^{-1}}$ for the updated inventory using peaks only (upd:elv) (see Table 2). A total of $1900 \pm 700$ TS5 $\mathrm{kg\,h^{-1}}$ of these emissions was attributed to a natural source spanning the whole modeling domain (potentially the Elbe River and associated wetlands).

For the municipal area of Hamburg (including the river, the port, and industrial and residential zones), the sum of

natural and anthropogenic $CH_4$ emissions estimated by this study ranges from $1500 \pm 1200$ to $1600 \pm 920\,\mathrm{kg\,h^{-1}}$. The $CH_4$ emissions from natural processes for the Hamburg area were estimated to be $730 \pm 270\,\mathrm{kg\,h^{-1}}$; thus, the emission from anthropogenic sources are estimated to be around $900 \pm 510\,\mathrm{kg\,h^{-1}}$ (for the original prior).

When we split anthropogenic emissions in Hamburg into biogenic emissions and emissions of thermogenic origin according to the split in the TNO GHGco inventory (see Table A1), $480 \pm 260\,\mathrm{kg\,h^{-1}}$ of the emissions is attributed to thermogenic sources and $420 \pm 240\,\mathrm{kg\,h^{-1}}$ is attributed to an anthropogenic biogenic origin, such as wastewater or landfills (see Fig. 10).

If we only look at the part of Hamburg that is located north of the Elbe, which was also studied by Maazallahi et al. (2020), our emission estimate is $420 \pm 230\,\mathrm{kg\,h^{-1}}$ for anthropogenic sources. This is higher than the $46 \pm 8.0\,\mathrm{kg\,h^{-1}}$ reported by Maazallahi et al. (2020) in their study based on upscaling emissions from a mobile $CH_4$ survey with a car. The difference can be partly explained by the different scientific objectives (and thus methodologies) used in both studies. While our study targeted total emission quantification (i.e., from all sources) using column instruments and, thus, can also capture sources that are emitting above the street level, the in situ measurements carried out by Maazallahi et al. (2020) were used to specifically target ground-level emissions near public roads, including the identification and quantification of fugitive emissions from gas pipeline leaks and the sewer system (not including the wastewater treatment plant). If we consider only fugitive emissions according to the TNO split (Table A1), our study estimates emissions of $210 \pm 110\,\mathrm{kg\,h^{-1}}$ for the northern part of Hamburg. This is between 2 and 8 times higher than the estimate presented by Maazallahi et al. (2020). One potential source that is usually not measurable at the street level, and could thus explain the lower emissions measured by Maazallahi et al. (2020), is end use inside homes (e.g., cook stoves and boilers for heating) (Lebel et al., 2022; Defratyka et al., 2021).

**Table 2.** Emission estimates for modeling domain. TS4

|  | Original | upd:all | upd:elv | Prior emissions |
|---|---|---|---|---|
| Domain | $6300 \pm 3500$ | $6300 \pm 4100$ | $5800 \pm 4500$ | 6600 |
| Natural (Domain) | $1900 \pm 700$ | $1800 \pm 680$ | $1800 \pm 700$ | 350 |
| City | $1600 \pm 920$ | $1600 \pm 1000$ | $1500 \pm 1200$ | 1500 |
| Anthropogenic (City) | $900 \pm 510$ | $860 \pm 560$ | $800 \pm 620$ | 1400 |
| Natural (City) | $730 \pm 270$ | $710 \pm 270$ | $710 \pm 280$ | 140 |

The emission estimates are reported in kilograms per hour for the different sections of the study area. "Domain" refers to the entire modeling domain including natural and anthropogenic sources. "City" refers to natural and anthropogenic emissions calculated for the area inside the municipal area of Hamburg. "Natural (Domain)" and "Natural (City)" refer to the emissions from natural sources in the whole modeling domain and in the city, respectively. "Anthropogenic (City)" refers to emissions from anthropogenic activity in the city area. Numbers in the table are shown with two significant digits.

Accumulated emissions from end use, while not affecting street-level concentrations, could be observable in total column measurements and, thus, contribute to the higher emission estimates of this study. Another source in Hamburg that could potentially contribute to higher column-measurement-based estimates is CE5 the Alster lakes near the city center. Around these lakes, Maazallahi et al. (2020) detected $CH_4$ enhancements that were low in magnitude but spread over a large area. These low enhancements could not be used for quantification and are, thus, not included in their estimate, but they might be noticeable in the column measurements.

### 3.6 Emission estimates from the mobile survey

For several locations inside the study area, emission estimates were derived using a GPM from transects recorded during the mobile survey. All transects for each location were undertaken on the same day. These estimates are presented in Table 3 and are compared to the emissions recorded in the TNO GHGco inventory as well as the two updated versions. While the emissions of the two updated inventory versions were only spatially redistributed according to the recorded spatial distribution of $CH_4$ concentrations, the GPM emission estimates consider wind information to obtain emission estimates.

For location 1, an oil refinery, sample bags were analyzed and an isotopic signature of thermogenic $CH_4$ emissions was detected. These emissions were quantified as $7.9 \pm 5.3 \, \text{kg h}^{-1}$ by driving multiple transects around the source location, as visualized in Fig. A5. This value is significantly larger than the value ($0.61 \, \text{kg h}^{-1}$) recorded in the TNO GHGco inventory for thermogenic $CH_4$ emissions in the corresponding pixel. Moreover, there is no source recorded in the European Pollutant Release and Transfer Register (E-PRTR; European Environment Agency, 2022), which suggests that it is an unknown source. The updated version of the inventory upd:all, with a value of $6.4 \, \text{kg h}^{-1}$, is closest to the Gaussian plume emission estimate for the corresponding inventory pixel. The updated inventory version upd:elv suggests even higher emissions for that pixel

($76 \, \text{kg h}^{-1}$). The fact that the source at location 1 was observed on several measurement days (and had also already been observed during the measurements in 2018) suggests that this source could have been emitting continuously for a longer time.

At location 2 north of the Elbe, the industrial area, and north of the municipal wastewater plant, transects were carried out, and an emission estimate was derived from the measured plumes, as shown in Fig. 9. This estimate of $6.7 \pm 13 \, \text{kg h}^{-1}$ has a high relative uncertainty because the estimated source location was far away from the transect lines. The GPM estimate is not significantly different from the values reported in the original and two updated inventory versions (5.4, 15, and $19 \, \text{kg h}^{-1}$, respectively). Emissions for this location were estimated during a period with southerly wind directions; thus, they could have originated from various sources within the industrial area as well as from the wastewater treatment plant. At this location, no samples were taken because plumes were not always stable.

Location 3 is situated in the industrial complex south of the Elbe near harbor water ways and adjoining several ports used to load or fill boats with gas- and oil-derived products (see Fig. 9). For this location, several large plumes were observed at the site of a refinery. These were attributed to thermogenic and biogenic source signatures. Biogenic sources, however, turned out to be dominant in a Keeling analysis of the sample bags. Biogenic emissions could have originated from near the waterbody or from the fermentation of wastewater from the facility. The estimated emissions from this location are $3.1 \pm 2.3 \, \text{kg h}^{-1}$, which confirms the value recorded in the corresponding TNO GHGco inventory pixel ($4.0 \, \text{kg h}^{-1}$). The updated inventory version "upd:all" ($4.6 \, \text{kg h}^{-1}$) is not significantly different from the GPM estimate, whereas the value in the upd:elv version is significantly higher ($40 \, \text{kg h}^{-1}$).

The transects at location 4 were undertaken on the private roads of a refinery after permission was granted from the operator. The first drives were distributed around the accessible area of the refinery, and they were then narrowed down to locations where plumes were detected. The emissions of a

**Table 3.** Emission estimates from mobile measurements.

| L | Lat | Long | Type | GPM (kg h$^{-1}$) | Original (kg h$^{-1}$) | upd:all (kg h$^{-1}$) | upd:elv. (kg h$^{-1}$) | Signature | Transects | Distance (m) |
|---|-----|------|------|------|----------|---------|----------|-----------|-----------|----------|
| 1 | 53.468 | 10.187 | Refinery | 7.9 ± 5.3 | 0.61 | 6.4 | 76 | t | 12 | 130 ± 17 |
| 2 | 53.539 | 9.943 | Undefined | 6.7 ± 13 | 5.4 | 15 | 19 | – | 6 | 720 ± 240 |
| 3 | 53.505 | 9.951 | Refinery | 3.1 ± 2.3 | 4.0 | 4.6 | 40 | b | 14 | 220 ± 55 |
| 4 | 53.483 | 9.969 | Refinery | 1.1 ± 0.7 | 1.5 | 1.3 | 1.8 | t | 11 | 180 ± 40 |
| 5 | 53.427 | 10.062 | Farm | 8.4 ± 2.5 | 0.87 | 0.55 | 3.8 | b | 6 | 310 ± 59 |
| 6 | 53.513 | 9.944 | Refinery | 6.6 ± 13 | 4.5 | 2.8 | 4.1 | – | 5 | 220 ± 190 |
| 7 | 53.505 | 9.948 | Refinery | 4.5 ± 4.4 | 4.9 | 3.0 | 4.4 | – | 4 | 470 ± 200 |

The emission estimates (GPM) from the mobile survey are reported in kilograms per hour for selected point source locations (L) in the study domain. Estimates are compared to the emissions recorded in the TNO GHGco inventory in the "Original" column (without natural emissions). The "upd:all" and "upd:elv" columns refer to the updated versions of the inventory (including natural emissions). The "Signature" column records the isotopic source signature type: t (thermogenic) or b (biogenic). Records marked with "–" were not analyzed for source type.

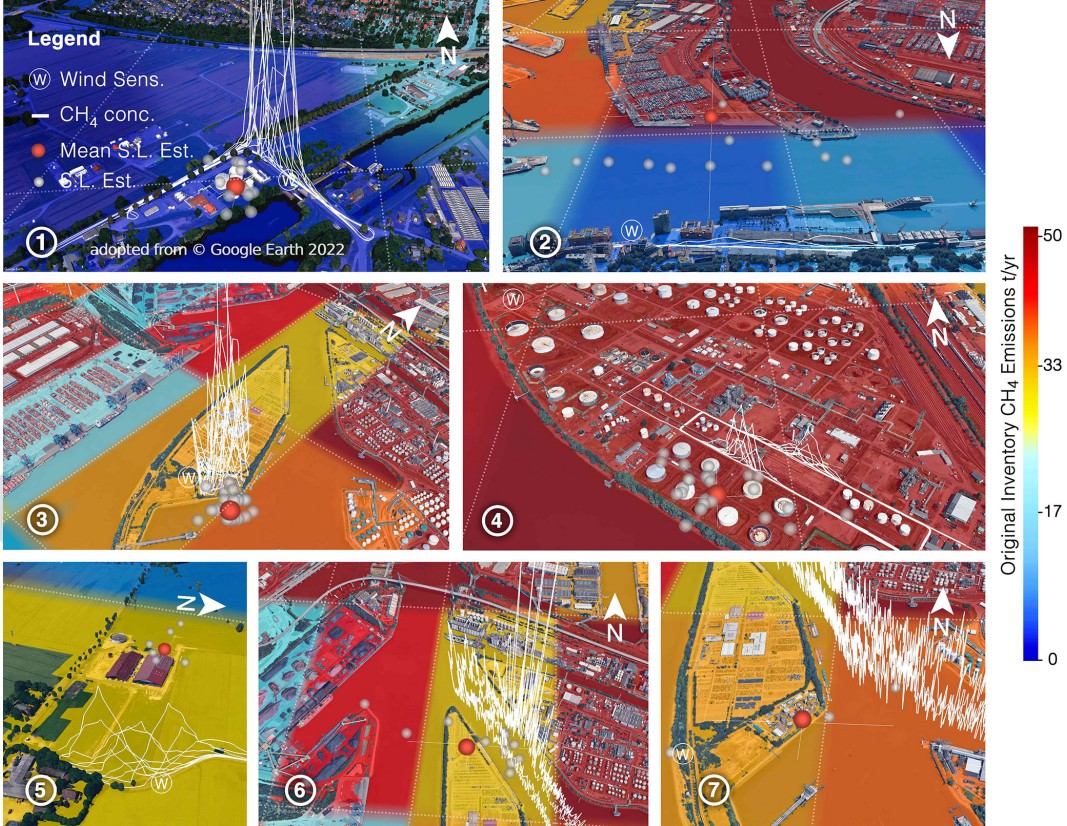

**Figure 9.** Transects of CH$_4$ concentration measurements are visualized using white lines. These were used to determine the emission strength of point sources with a Gaussian plume model (GPM). Estimated source locations (S.L. Est.) are shown using gray spheres. The mean source location estimate (Mean S.L. Est.) is shown using a red sphere, with white perpendicular lines indicating the error in the Mean S.L. Est. The background colors indicate the emissions recorded in the original TNO GHGco inventory (biogenic and thermogenic). Inventory pixels are separated by a white dotted line, and they have an approximate length of 1100 m and a width of 650 m at this latitude. Blue areas indicate zones where the original inventory has low emissions recorded, whereas red and yellow areas indicate high-emission zones. The locations where a local wind sensor has been mounted are marked with a "W". Location 6 used the same wind sensor as location 7. Images were taken from Google Earth.

prominent point source, present during the time of the survey, were quantified as being $1.1 \pm 0.7\,\mathrm{kg\,h^{-1}}$, which confirms the values recorded in the TNO GHGco inventory and the updated versions (as can be seen in Table 3).

At location 5, plumes were detected downwind of two large sheds situated on a farm near Meckelfeld. Isotope measurements of air samples collected at this location indicated a biogenic source origin. For this source, the upd:elv inventory provides the closest estimate of $3.8\,\mathrm{kg\,h^{-1}}$. The GPM estimate of $8.4 \pm 2.5\,\mathrm{kg\,h^{-1}}$ is considerably higher than the values recorded in the original and the upd:all version of the inventory (0.55 and $0.87\,\mathrm{kg\,h^{-1}}$, respectively).

Transects at locations 6 and 7 were both carried out by boat. Two point sources with respective magnitudes of $6.6 \pm 13$ and $4.5 \pm 4.4\,\mathrm{kg\,h^{-1}}$ were found in the industrial area. No sample bags were analyzed for these locations. For both locations, the original inventory is closest to the emission estimate; however, the difference between the updated and original inventories is small. Both estimates have a high relative uncertainty, as only very few transects were available and the estimated source location could possibly be too far from the transects. Both GPM estimates are not significantly different from the values recorded in the original and updated inventory versions.

In general, several significant $CH_4$ sources were quantified during the mobile survey. While several GPM estimates confirmed the values recorded in the emission inventory (both the updated and original versions), some of the biogenic and thermogenic sources estimated using GPM, like locations 1 and 5, were significantly above the values recorded in the TNO GHGco inventory. The correlation between GPM estimates and the inventory values is highest for the upd:elv inventory: $R^2 = 0.13$ compared with $R^2 = 0.10$ and $R^2 = 0.10$ for the upd:all and the original inventory, respectively. On the other hand, the root-mean-square error (RMSE) is highest for the upd:elv inventory: $27\,\mathrm{kg\,h^{-1}}$ compared with the upd:all and the original inventory with a RMSE of 4.4 and $4.1\,\mathrm{kg\,h^{-1}}$, respectively.

## 3.7 Comparison with other emission inventories

The emissions from anthropogenic activity in the city of Hamburg were estimated to be $900 \pm 510\,\mathrm{kg\,h^{-1}}$, which is not significantly different from the $1400\,\mathrm{kg\,h^{-1}}$ reported in the TNO GHGco inventory for the year 2015.

During our study, we observed influence from a biogenic source, which was modeled as river emissions. Large natural area sources such as waterbodies were previously not recorded in the TNO GHGco inventory.

The column-measurement-based $CH_4$ emission estimates for all sectors (natural and anthropogenic sources) in the whole domain, covering the city of Hamburg and parts of the surrounding land outside of Hamburg, are of the same magnitude as those reported by inventories, as can be seen in Fig. 10.

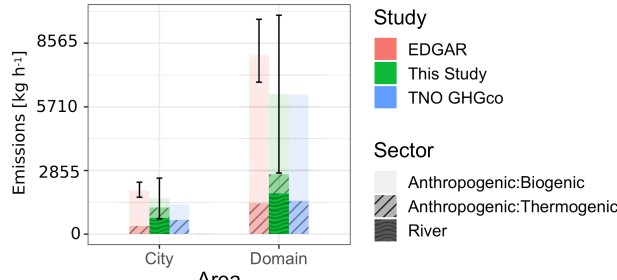

**Figure 10.** Comparison of inventories and the emission estimates of this study for the city of Hamburg and the whole modeling domain. Emission estimates are split by emission sector according to the split in the TNO GHGco inventory. Error bars for EDGAR (Emissions Database for Global Atmospheric Research) are the overall uncertainties for EDGAR GHGs from Solazzo et al. (2021). For the TNO GHGco inventory, no uncertainty is available for $CH_4$. The TNO GHGco and the EDGAR inventory both do not include river emissions.

## 3.8 Isotope measurements

The stationary in situ measurements on the rooftop of the Geomatikum building (University of Hamburg) show numerous concentration peaks with enhancements of around 1–2 ppm, as visible in Figs. 11 and A1. During the campaign, these peaks were only measured during the night or when the column instrument was not measuring due to cloud cover. Both $\delta^{13}C$ and $\delta D$ Keeling plots yield source signatures that indicate a biogenic origin ($\delta^{13}C$ $-61.5\,\permil \pm 0.3\,\permil$ and $\delta D = -320\,\permil \pm 2.5\,\permil$) for these peaks, as can be seen in Fig. 12. Potential sources that generally have a similar signature are microbial in nature (Menoud et al., 2021). Both agricultural sources, such as cattle (Lu et al., 2021), and waste have overlapping signatures with the unknown source in Hamburg. Dietrich et al. (2023) found a similar signature ($\delta^{13}C$ $-66.1\,\permil$ and $\delta D = -310\,\permil$) for air in a subway station. A study on a river estuary at the border between Belgium and the Netherlands by Jacques et al. (2021) found a comparable signature for $\delta^{13}C$ (between $-25.2\,\permil$ and $-65.6\,\permil$) but a more enriched signature for $\delta D$ (between $+101\,\permil$ and $-212\,\permil$). $\delta D$ signatures of as low as $-260\,\permil$ have been measured by Martens et al. (1999) for gassy sediments in an estuary in Germany. The slightly more depleted $\delta D$ signature measured in this study suggests that the unknown source in Hamburg could be a mix of several different biogenic (microbial) sources. One of these could be a large natural $CH_4$ source, such as the river or wetlands, that emits in Hamburg (see also Sect. 3.2). However, the river flow in the city area is also influenced by anthropogenic activity (e.g., harbor traffic and wastewater) which could contribute to lower $\delta D$ values. The sharp short-term peaks could be caused by canals in the city close to the in situ instrument; these fall dry during low tide and then fill up again during high tide. This hypothesis is supported by the tempo-

ral correlation of $CH_4$ peaks with the rising tide, as visible in Fig. 11. Furthermore, less-pronounced peaks, such as those in the early morning on 20 August, 31 August, and 2 September, follow this pattern.

## 4   Discussion

In our study, two ways of correcting the spatial distribution of the prior emission map were attempted: (1) the addition of sources quantified by other studies that are not yet part of standard inventories, such as the TNO GHGco inventory (river emissions were previously reported by Matousu et al., 2017), and (2) the correction of the spatial distribution of existing gridded inventories via mobile measurements.

The example of 31 August illustrates how the first approach can have a significant impact on the modeling. When a localized source is not in the inventory but is observable in the measurements, the framework cannot model the prior expected concentrations correctly and, thus, the modeled enhancement is inexplicably low. In this case, the inversion framework will adjust the background to higher values than the measurements and, thus, can lead to negative enhancements as well as negative emissions. Only when the spatial distribution of the emission sources in the model is representative of the real distribution is the inversion framework able to constrain the total emissions based on the measurements. Once the river was added as a new source to the emission map, the results turned from negative emissions to positive emissions, which shows that adding unlisted sources to the inventory can improve the modeling significantly. Alternative reasons for the observed behavior could also be an overly low prior uncertainty or sources outside of the domain.

The results for 31 August suggest higher emissions from a source north of the West (mc) station. In this paper, the source was modeled as river emissions, but it could also be caused by another source further north of the Elbe or outside of the modeling domain. For instance, if there were large cattle farms to the north of West (mc), these could possibly produce similar enhancements and would also match the isotopic signature measured in this study. During the campaign, however, no mobile survey was conducted north of West (mc) and the river that could have revealed emissions from the agricultural sector. Moreover, exceptional emissions from ships circulating on the river could cause or contribute to similar enhancements. Other studies in urban environments, such as Pickard et al. (2021), found that polluted urban lakes in India contribute significantly to $CH_4$ and $CO_2$ emissions. Furthermore, a study by Zazzeri et al. (2017), who measured isotopic $CH_4$ signatures in London, suggested that river emissions can contribute significantly to the $CH_4$ mix. The proximity to the shelf areas of the North Sea and the Elbe Estuary could additionally influence the measurements, as around 75 % of ocean $CH_4$ emissions come from these areas (Bange et al., 1994). For natural sources, like a river, an oscillation

of emissions with the tide cycle could be expected. However, such oscillations could not be resolved in our daily emission estimates derived from column measurements. In contrast, the analysis of the isotope in situ data showed such a correlation with the rising tide, suggesting that the peaks could be caused by the river and its connected waterbodies.

In the future, the inversion framework should be developed further to include in situ $CH_4$ concentration data along with column concentrations. This way, the modeling could be improved and the inversion could further constrain the emission estimates as well as providing more insights into whether river emissions could in fact explain the observed enhancements.

Other potential sources include $CH_4$ emissions from soft soil layers, as reported by the city administration for the Elbe glacial valley, which is located mainly to the south of the current river course (Hummel and Eickers, 2022). While rather unlikely during a day with moderate wind speeds, these emissions from the ground could have accumulated near the instrument location and caused the observed rise in concentration.

The isotopic signals observed in Hamburg are probably a mix of several microbial sources of natural and anthropogenic origin. Further investigations are necessary here, and mobile measurements near the wetland, which was covered by the measurement footprints on 31 August, could provide better insight.

Furthermore, the second approach, the correction of the spatial distribution of sources with mobile measurements, has an effect, especially on individual days. This may be due to temporal variability in the emissions (different sources emit only for a short period of time); thus, sometimes the updated inventory matches better, whereas the original inventory can be used to more accurately model the observed enhancements on other days. In addition, on different days, due to specific wind directions, different sections of the inventory are covered by the measurement footprints. In some sections, the differences between the original and the updated inventories are more pronounced than in other regions. Moreover, it is possible that one of the principal assumptions of the framework – that the background concentration of the whole domain boundary is equal at each time stamp – does not always hold.

However, the average emission estimate "all dates" remains relatively constant for the three versions of the emission inventory. This indicates that the result is representative of total city emissions when averaging over multiple measurement days, and variations in the spatial distribution of prior emissions are of minor importance, although they can be important for single days due to the reasons mentioned above. The variability among individual days is quite large, which could also indicate the limits of the Bayesian inversion for short measurement periods.

The correction of the inventory using mobile measurements seems to be a promising approach to update the spatial

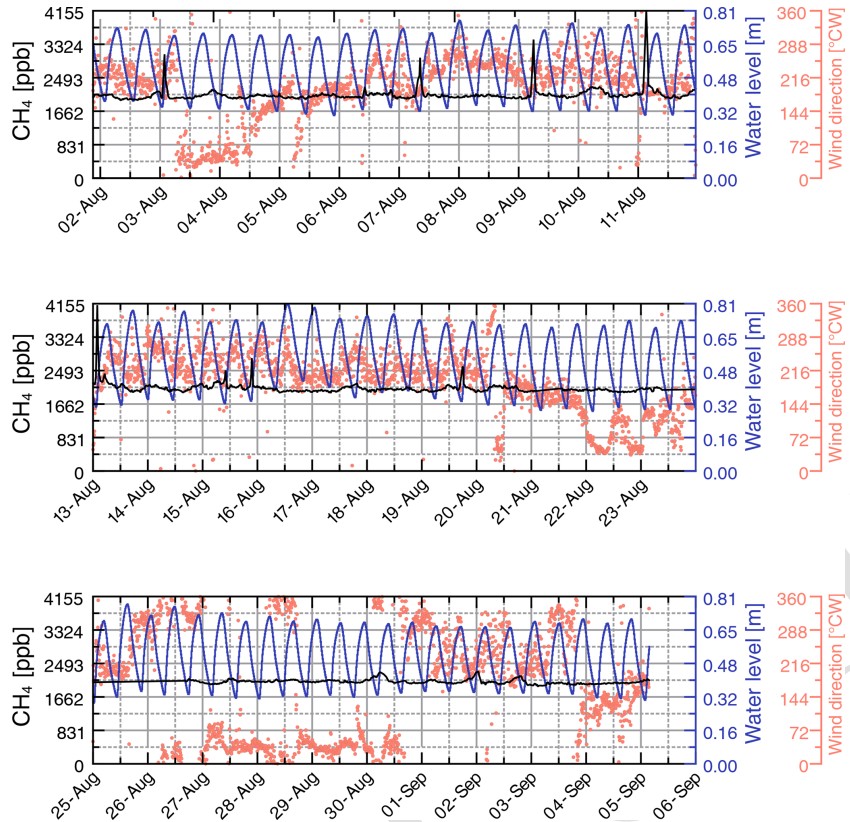

**Figure 11.** In situ $CH_4$ and wind direction time series from the rooftop of the Geomatikum building, Hamburg. A correlation of the measured peaks with the tide cycle is visible. Water level data from Bundesanstalt für Gewässerkunde (2021).

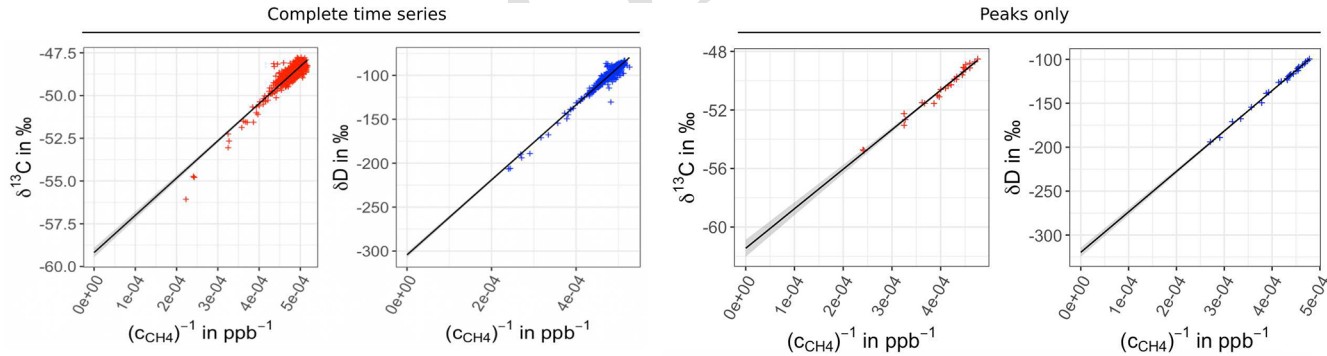

**Figure 12.** Keeling plots for C and H isotopes for the complete time series (left panel pair) and for all of the $CH_4$ peaks during the campaign period (right panel pair). The complete time series signatures were $\delta^{13}C = -58.9 \pm 0.2\,‰$ and $\delta D = 306\ \text{TS6} \pm 1.5\,‰$; the peak-only signatures were $\delta^{13}C = -61.5\,‰ \pm 0.3\,‰$ and $\delta D = -320\,‰ \pm 2.5\,‰$.

distribution of emissions. However, mobile measurements cannot be carried out everywhere at once, and multiple drives over the course of weeks need to be combined to obtain corrections at the city scale. The representativeness of this relatively short snapshot of the measured concentration of the yearly emissions needs to be studied further. Nevertheless, it should provide a better estimate than bottom-up inventories in some cases and could be used to distribute emissions

on higher-resolution grids in areas where there are no high-resolution inventories available.

The combination of the two correction methods – the inclusion of natural sources and the use of mobile measurements – can improve the spatial distribution of the prior emission map. Scaling this updated map according to the findings of an inversion framework (using column concentration measurements) turns out to be a feasible technique to update city-

scale emission inventories. To yield representative emission inventories, however, this approach would need to be carried out for a longer time period than that employed in the present study.

In this study, we have updated all sectors of the emission inventory at once. However, as mobile measurements are only sensitive to near-ground sources, such as fugitive emissions from gas infrastructure and wastewater, the information obtained from mobile surveys could only be used to correct the corresponding sectors in the inventory in future studies.

In order to improve the inversion framework, further work is necessary, especially regarding approaches on how to find a more reliable background prior. At the moment, a constant value has been used that is then fitted by the framework to the measurements. This can lead to errors, especially when the spatial and temporal variations in the emissions in the inventory do not conform to the measured enhancements.

The emission estimate for the city of Hamburg was derived over a period of 1.5 months, and the GPM estimates were derived during an even shorter time and, according to Brantley et al. (2014), might not be representative of yearly emissions. Long-term measurements, especially in the different seasons of the year, are necessary to quantify the quite variable ensemble of sources. Furthermore, the prior emission inventory is based on average yearly emissions (summer and winter months); thus, the prior emissions could not be fully representative of the study period in the summer.

Natural sources, such as the river, might be emitting more in the summer, while natural-gas-fired heating is mainly used in the winter months. The gap between the emission estimate of the mobile survey by Maazallahi et al. (2020) and the column-based estimate derived in this study could, in the future, be investigated further. For instance, measuring indoor fugitive emissions in representative households and upscaling these results to the city scale could provide further insights into where the difference is coming from.

During the mobile surveys, we visited several refineries in the harbor area. One large refinery was in the process of disassembly, as the industrial site is moving to another location in Germany. This example shows that, although the measured emissions are currently lower than the emission inventory suggests, sources such as industrial processing sites might have just moved their facilities and are now emitting somewhere else. Thus, further studies and updated emission inventories that consider the spatial changes in emission sources over time and across administrative borders and countries are necessary.

## 5 Conclusions

This study shows the challenges of quantifying $CH_4$ emissions of a large source region like the municipal area of Hamburg. The approach using FTIR spectrometers and a

Bayesian inversion framework turned out to be dependent on the correct modeling of the emission sources in the prior emission inventory. The addition of river emissions, which were quantified in a previous study by Matousu et al. (2017), was necessary to obtain positive emission estimates on 31 August. Small sources and sectors could not be quantified separately using this methodology, as the expected concentrations were below the instrument precision. The emission estimate derived in this study has a large uncertainty, and estimates from the bottom-up TNO GHGco and EDGAR inventories are not significantly different. Further good measurement days distributed throughout a year would be needed to obtain a more certain estimate. Moreover, further improvements to the small-domain inversion system could be made to exclude the possibility of the boundary conditions affecting the emission estimates. Our study shows that it is feasible to correct the spatial distribution and the magnitude of sources in emission inventories using a combination of mobile measurements and the inversion of column measurements. The addition of natural sources that were not listed in the inventory improved the modeling significantly on some days. Furthermore, the corrections using mobile measurements changed the emission estimates for particular days, and this effect averaged out for the whole campaign period; the "all dates" estimate was similar for updated and non-updated inventories. On the one hand, our analysis of column measurements suggests that there is a large natural $CH_4$ source, potentially the Elbe River, in Hamburg that is not listed in common emission inventories. Some standard inventories, such as the TNO GHGco inventory, do not include natural sources (e.g., wetlands and rivers) and adding these manually to the inventory can improve the modeling. On the other hand, our isotope measurements revealed $CH_4$ signals that were attributed to a biogenic origin. The timing of the measured $CH_4$ peaks correlates with the rising tide in the river estuary, which makes a connection between the observed peaks and the river system more likely. Further investigations are necessary to establish if this source is in fact the Elbe River and wetlands or if the calculated natural emissions are a summation of several independent biogenic sources (of natural and anthropogenic origin). The isotope measurements in Hamburg were continued until 28 March 2022, and a future study will provide more insights into this in the near future.

Although the contributions from natural sources are significant in Hamburg ($730 \pm 270$ TS7 kg h$^{-1}$), the study also shows that the largest share of total $CH_4$ emissions in Hamburg are of anthropogenic origin ($900 \pm 510$ kg h$^{-1}$). A comparison between an earlier study in Hamburg (Maazallahi et al., 2020) and this study showed that the $CH_4$ emissions derived via street-level mobile measurements could potentially underestimate total emissions, as they do not capturing natural-gas-related $CH_4$ emissions from end use in homes (e.g., gas stoves and boilers for heating; Lebel et al., 2022; Defratyka et al., 2021; Dietrich et al., 2023). Furthermore,

large area sources, such as the Alster lakes or the Elbe, could contribute to the differences in emission estimates. In the course of this study, a large and, thus far, unknown emission source of thermogenic origin was located at a refinery and was quantified, using mobile measurements, to be $7.9 \pm 5.3\,\mathrm{kg\,h^{-1}}$. This finding highlights the need for further surveys of unknown sources in cities and that an increased effort with respect to the reduction of anthropogenic $CH_4$ emissions in cities is required.

# Appendix A

## A1 Figures

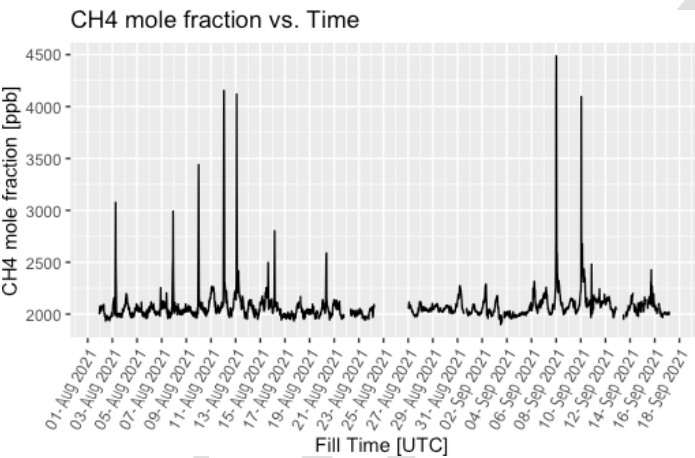

**Figure A1.** Stationary in situ measurements of $CH_4$ for a longer time frame. Peaks are even visible after the end of the campaign. These will be discussed in more detail in a future study.

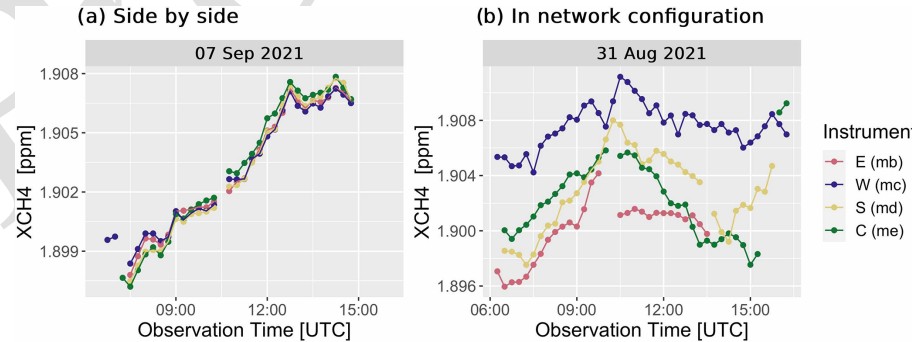

**Figure A2.** Measurements of the four FTIR instruments after calibration: **(a)** co-located (side-by-side) instruments on the rooftop of the Geomatikum, Hamburg (mismatch between instruments of $0.21 \pm 0.48\,\mathrm{ppb}$); **(b)** instruments in a network configuration according to Fig. 1.

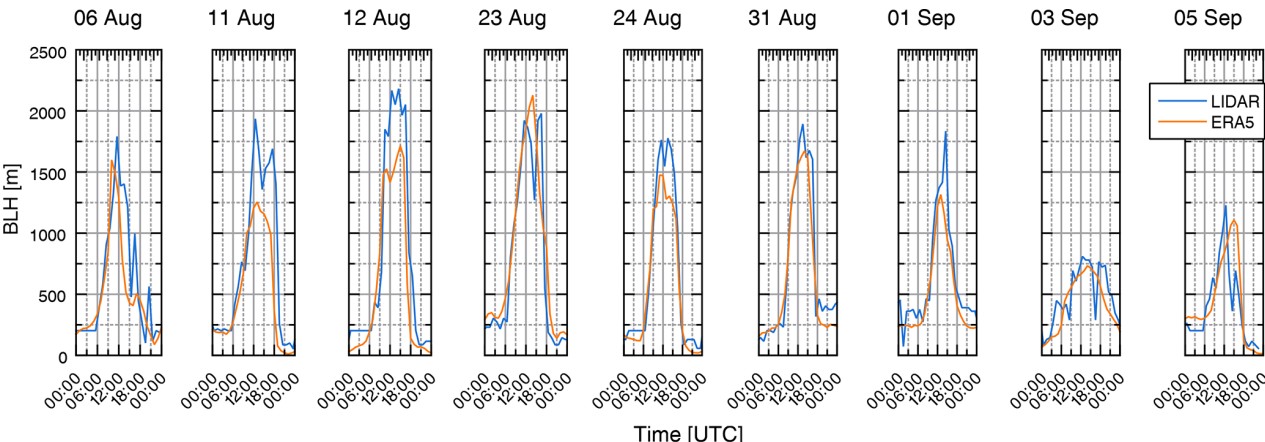

**Figure A3.** Planetary boundary layer height comparison: the estimate extracted from lidar turbulence measurements vs. the ERA5 model result. For the campaign period, good agreement was found between the model and lidar results.

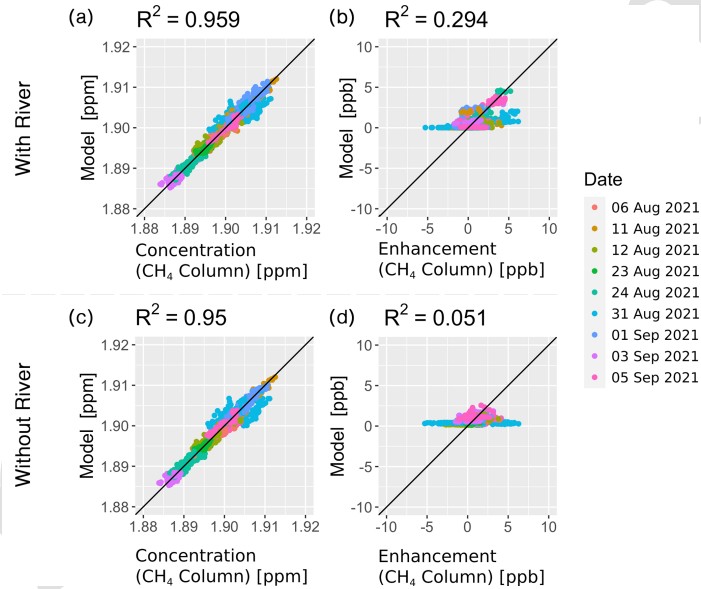

**Figure A4.** Regression plot of the measured and modeled $CH_4$ signal for all 9 selected measurement days. Panels **(a)** and **(b)** show the result for a prior with the river added as a separate sector. Panels **(c)** and **(d)** show the result using the unchanged TNO GHGco inventory (no river emissions added). Panels **(a)** and **(c)** refer to the whole signal (background and enhancement), whereas panels **(b)** and **(d)** show the correlation for the enhancement only. The addition of the river emissions increased the correlation of the enhancement significantly.

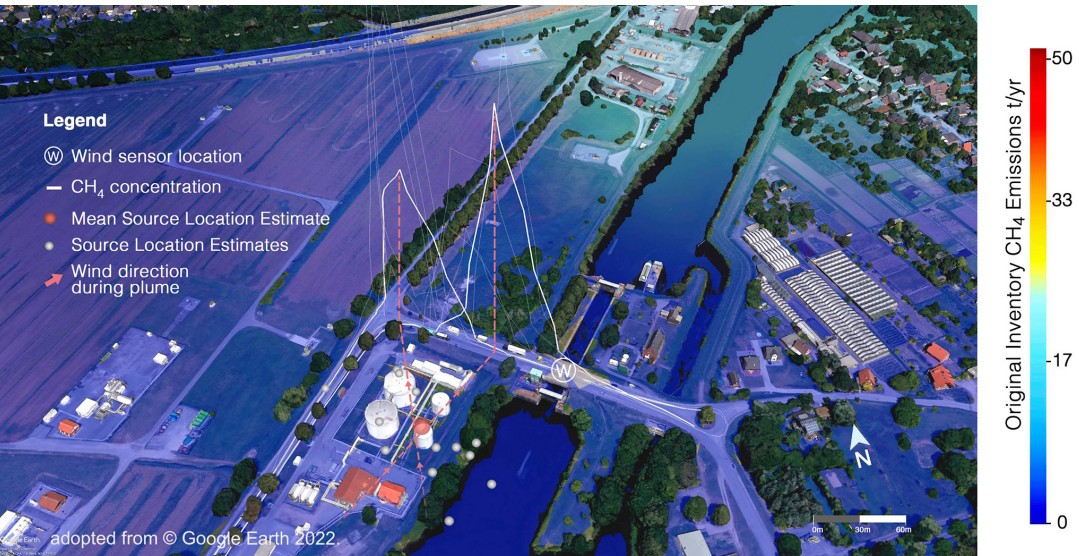

**Figure A5.** Visualization of the mobile measurements around an oil refinery (location 1) near Bergedorf, Hamburg. The measured CH$_4$ plumes are shown using white lines. Two distinct plumes are highlighted for slightly different wind directions. For all recorded plume transects, a source location estimate has been derived (gray spheres). The mean estimate for the source location is shown as a red sphere that is co-located with one of the refinery tanks. The background colors indicate the emissions recorded in the original TNO GHGco inventory. Blue areas indicate zones where the original inventory has low emissions recorded.

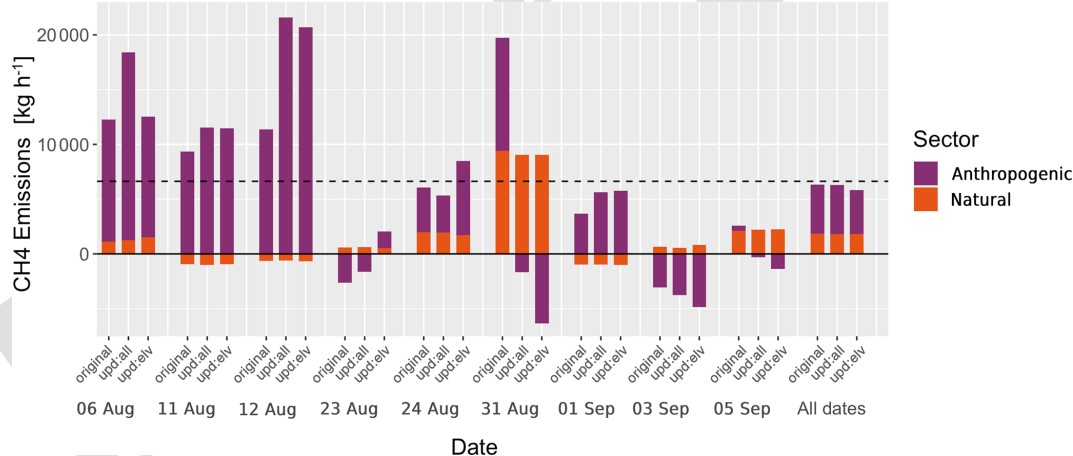

**Figure A6.** The results of the inversion split by the two sectors (river and anthropogenic) used in the modeling.

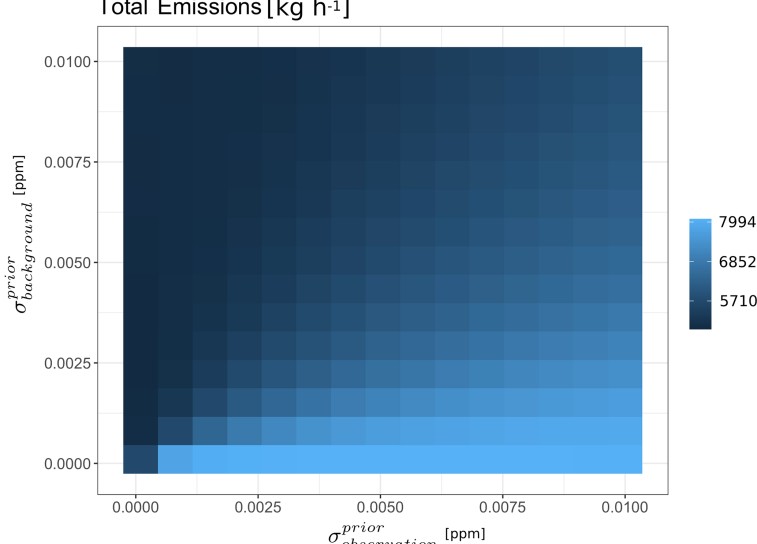

**Figure A7.** The two inversion parameters $\sigma_{\text{observation}}^{\text{prior}}$ and $\sigma_{\text{background}}^{\text{prior}}$, which represent the uncertainties in the observations and the background estimate, respectively, have been varied systematically. The final emissions for the whole domain are shown for each parameter combination in this plot. Emissions are quite stable for all realistic parameter combinations.

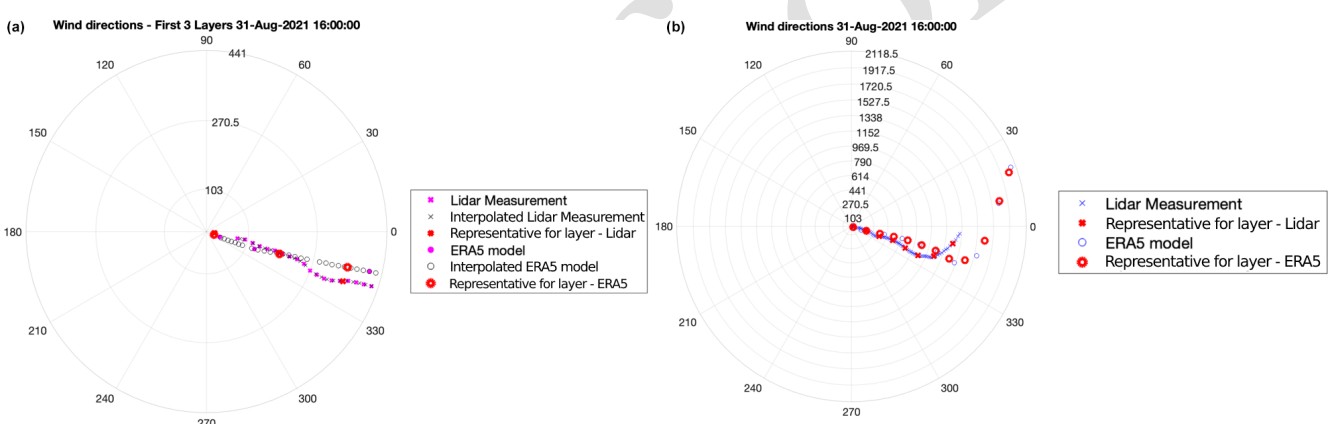

**Figure A8.** Comparison of lidar wind data and ERA5 model data on 31 August. The circles in the wind rose plot correspond to different altitudes (in meters). The wind direction (in degrees) is plotted for each height level. Panel **(a)** is a zoomed-in version of panel **(b)** and shows how measurement and model results were interpolated for the comparison. In panel **(a)**, raw data are represented by the color magenta, interpolated data are represented by the color black, and the representative value of each height layer is represented by the color red. The angular distance between the red circles and red crosses in each plot corresponds to the wind direction mismatch in each layer. Panel **(b)** shows all of the height levels used to compute the mismatch.

## A2  Tables

**Table A1.** Assignment of inventory sectors to the biogenic and thermogenic categories.

| Inventory | Thermogenic | | Biogenic | |
|---|---|---|---|---|
| | Abbreviation | Description | Abbreviation | Description |
| EDGAR | CHE | Chemical processes | AGS | Agricultural soils |
| | ENE | Power industry | AWB | Agricultural waste burning |
| | FFF | Fossil fuel fires | ENF | Enteric fermentation |
| | IND | Combustion for manufacturing | MNM | Manure management |
| | IRO | Iron and steel production | SWD | Solid waste |
| | PRO | Fuel exploitation | | |
| | RCO | Energy for buildings | WWT | Wastewater handling |
| | REF TRF | Oil refineries and transformation industry | | |
| | TNR | Aviation, shipping and railway | | |
| | TRO | Road transportation | | |
| TNO | A | Public power | K | Agricultural livestock |
| | B | Industry | L | Agricultural other |
| | C | Other stationary combustion | J | Waste |
| | D | Fugitive emissions | | |
| | E | Solvents | | |
| | F1–3 | Road transportation | | |
| | G | Shipping | | |
| | H | Aviation | | |
| | I | Off-road | | |

Classification of the different source sectors in biogenic and thermogenic emissions in the TNO and EDGAR inventories. Given that the TNO inventory does not separate waste into subcategories, we treated all of the sources from waste in the EDGAR inventory as one for consistency.

**Table A2.** Transport error in parts per million.

| Date | upd:elv | upd:all | Original |
|---|---|---|---|
| 6 August 2021 | 0.00072 | 0.00062 | 0.00066 |
| 11 August 2021 | 0.00071 | 0.00068 | 0.00070 |
| 12 August 2021 | 0.00056 | 0.00053 | 0.00053 |
| 23 August 2021 | 0.00056 | 0.00053 | 0.00056 |
| 24 August 2021 | 0.00074 | 0.00068 | 0.00082 |
| 31 August 2021 | 0.00073 | 0.00074 | 0.00073 |
| 1 September 2021 | 0.00098 | 0.00101 | 0.00105 |
| 3 September 2021 | 0.00078 | 0.00066 | 0.00075 |
| 5 September 2021 | 0.00119 | 0.00101 | 0.00110 |

Average transport error in parts per million as calculated for each day and each of the three inventories ("upd:all", "upd:elv", and the original TNO GHGco inventory) by rotating the trajectories of the particle files according to the standard deviation of the lidar vs. ERA5 model mismatch.

**Table A3.** Average ERA5 and lidar wind data.

| | Mean wind speed (m s$^{-1}$) | | Mean wind direction (° CW) | |
|---|---|---|---|---|
| Date | Lidar | Model | Lidar | Model |
| 6 August 2021 | 6.1 | 5.0 | 158 | 160 |
| 11 August 2021 | 4.1 | 4.1 | 273 | 261 |
| 12 August 2021 | 4.2 | 4.2 | 192 | 197 |
| 23 August 2021 | 7.5 | 6.8 | 55 | 49 |
| 24 August 2021 | 4.0 | 3.9 | 73 | 60 |
| 31 August 2021 | 4.6 | 4.6 | 8 | 354 |
| 1 September 2021 | 5.4 | 4.3 | 312 | 314 |
| 3 September 2021 | 6.3 | 5.1 | 298 | 291 |
| 5 September 2021 | 2.8 | 2.5 | 102 | 89 |

Daily mean wind speed and wind direction (model and lidar data) for selected campaign days that were used to estimate emissions.

**Data availability.** The retrieved $CH_4$ concentration measurements can be accessed at https://retrieval.esm.ei.tum.de/ (Makowski et al., 2023). The raw data can be provided by the corresponding authors upon reasonable request. The water-level data for the Elbe river can be obtained from https://www.pegelonline.wsv.de/webservices/files/Wasserstand+Rohdaten/ELBE/HAMBURG+ST.+PAULI (Bundesanstalt für Gewässerkunde, 2021).

**Author contributions.** JC, AF, and FD designed the concept and organized the campaign. AF, FD, JB, JC, HM, CS, and CvdV carried out the measurements. AF, FD, JC, DW, JB, and StS contributed to writing the paper. AF, JB, FD, DW, JC, HM, CS, MM, XZ, AU, FK, and HDvdG analyzed the data. TJ created the inversion framework and provided support. JC and FD acquired funding. JC supervised the project. TR and NW provided extra funding and instruments.

**Competing interests.** The contact author has declared that none of the authors has any competing interests.

**Acknowledgements.** The authors wish to acknowledge Felix Ament and Ingo Lange, who provided local expertise, support, and additional data. We are also grateful to Jan-Claas Böhmke, Frank Becker, Tobias Tiedgen, Dennis Fricke, Wolfgang Regge, Björn Brügmann, Rainer Knut, and Friedhelm Jansen for providing sensor locations and for their help with the measurements. Moreover, we are grateful to Andreas Luther, Vigneshkumar Balamurugan, and Haoyue Tang for their support with the lidar data.

Authors from the Technical University of Munich are grateful to Stefan Schwietzke and Daniel Zavala-Araiza for helpful conversation in their role as part of the Office of the Chief Scientist of the Climate and Clean Air Coalition Methane Science Studies (MSS), which is funded by the Environmental Defense Fund, the European Commission, the companies of the Oil and Gas Climate Initiative, and the United Nations Environment Programme (UNEP). Authors from the Technical University of Munich are additionally grateful for invitations to participate in workshops hosted by UNEP in the context of the IMEO Methane Science Studies.

This work was supported by the Climate and Clean Air Coalition (CCAC) Oil and Gas Methane Science Studies (MSS) hosted by the United Nations Environment Programme (reference no. DTIE20-EN1345).

Funding was provided by the Environmental Defense Fund, the Oil and Gas Climate Initiative, the European Commission, and CCAC. This research has further been supported by the Deutsche Forschungsgemeinschaft (DFG, German Research Foundation; grant nos. CH 1792/2-1 and INST 95/1544). Jia Chen is supported by the Technical University of Munich – Institute for Advanced Study, funded by the German Excellence Initiative and the European Union Seventh Framework Programme under grant agreement no. 291763.

**Financial support.** This research has been supported by the United Nations (grant no. DTIE20-EN1345), the Deutsche Forschungsgemeinschaft (grant nos. CH 1792/2-1 and INST 95/1544), and the Institute for Advanced Study, Technische Universität München (grant no. 291763).

This work was supported by the Technical University of Munich (TUM) in the framework of the Open Access Publishing Program.

**Review statement.** This paper was edited by Thomas Karl and reviewed by two anonymous referees.

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

## Remarks from the language copy-editor

## Remarks from the typesetter