# Peer review of "Quantification of methane emissions in Hamburg using a network of FTIR spectrometers and an inverse modeling approach"

_Atmospheric Chemistry and Physics, 2022_

## Author Response (AR1)

**Author's response**

Dear Thomas Karl, we have gone over the referee comments and have updated the manuscript accordingly. Please fnd our point-by-point replies in red in the text below. Please let us know if there are further changes required.

**List of relevant changes:**

- Equation 1-4 were added to make clearer how the prior inventory was updated using mobile measurements.
- Figure 2 was updated with a bigger map
- Equations of the isotopic analysis of isotopes were removed, according to a reviewers suggestion.
- Figure 4 was updated to include wind information
- Figure 8 was changed to show the correlation of the enhancement in more detail.
- Figure 11 was added to show the correlation between the tide cycle in the river estuary and the measured CH4 enhancements.
- Figure 12 was separated in panels for the peaks only and the complete signal, according to a reviewers suggestion.
- The conclusion was adapted to reflect the uncertainties in the inversion result.
- Figure A2 was added to show the side by side calibration of spectrometers.
- Figure A3 was added to show the agreement of measured and modeled planetary boundary height.
- Units throughout the manuscript were changed from Gg per year to kg per hour.
- For further detailed changes please refer to the point by point replies to the referee comments below.

**Reply to Referee Comment 1**

**General**

**Thank you for your detailed analysis of our manuscript and the general suggestions. We have incorporated your feedback into the manuscript and reply in line with your comments. Please find our replies in bold below**

*The paper presents the estimation of urban CH4 emissions in Hamburg using 4 total column spectrometers and an urban scale inversion framework. Mobile measurements were performed to improve the spatial distribution in the prior inventory. Compatibility of the measurements with added emission from the Elbe river is investigated. Strong efforts have been implemented to deploy a temporary network (1.5 months).*

*The approach and method is fundamentaly similar to Jones et al., ACP 2021 with a few refinements. The novelty includes essentially accounting for river emission. However, Jones et al., (2021) stated: "This shows that a single day of measurements is unlikely to produce a robust result; even the combination of all 5 d should be viewed as an experimental proof of concept, and continuous measurements for many months should be used to produce a more compelling final emission number. ». Here, 9 useful days led to uncertainties that still remain unable to meaningfully confront inventories. It is not clear from this small sample how many days would be required to significantly improve uncertainties? Beyond the interesting (but, I would argue, not fully conclusive) discussion on the river, the present research does not bring strong constraints on city CH4 emissions. I can imagine plenty of reasons why the authors were not able to measure for a longer period, as these campaigns are very demanding, but more days would be clearly needed.*

*The Discussion section critically examines the results.* *The authors could 'soften' statements throughout the paper to better align with their own reflections on the method limitations.*

**We have softened statements where required and added further evidence that supports the river hypothesis.**

*While the South, East and West EM27 seem to be located on the outskirt of the city and hence would be able to "encapsulate" emissions, the North sensor seems to be located amidst emissions (Line 120). As a result, the network seems to be challenging itself when dealing with north-south winds.* *What has led to this choice? I would imagine that this contributes to the strange behaviour of the inversion during event of Northern wind (31/8).*

**We have clarified our reasoning as follows in Line 122f: "The northern site was co-located with the isotope measurements on top of the roof of the Geomatikum building of the University of Hamburg. This location was chosen weighting different criterions firstly the availability of sites with suitable conditions to house the room-size setup for isotopic measurements as well as allowing for the set up of the FTIR instruments on top of a flat roof. The Second consideration was to place the site outside of the industrial area, a high emission zone according to the TNO GHGco inventory."**

*Given the limited spatial extent of the network and inversion domain, particles get out of the domain within 1-2h. As total column measurements, only a small part of the signal is linked to the boundary layer. I would assume that the background has a much larger influence than the emission convolved with footprint over the domain, and the background is poorly known. Would it be possible to demonstrate that the sensitivity to within-domain fluxes (Aas in Jones et al., 2021) is significant compared to background sensitivity (Bbs)?*

**We have changed Figure 8 to show that when separating the modelled enhancement into 0.5ppb bins, the means of all bins with enhancements higher than 0.5ppb are significantly (p=0.001) distinct from the 0-0.5ppb bin. This shows the frameworks ability to detect diffuse emissions.**

*Observation uncertainties may be optimistic and drive the inversion into unrealistic behaviour forcing to stick closely to observations even though the signals are small. Frey et al. (2019) suggest a 1-sigma precision comparison of 4.3 ppb, which could be an interesting option for uncertainty (accounting for biases), rather than the 0.2ppb @10 min from Chen et al., (2016) related to differential column measurements. Also, showing explicitly and using the results of the spectrometers' comparisons that have been done by the team (only briefly mentioned L111-113) would be useful to keep this uncertainty value in check. The isotopic measurements are not included in the inversion frameworks, which limits the impact of these valuable data to the entire discussion. It is unclear what really drives the isotopic signature found at the rooftop measurements.*

*Several statements are made with vague formulation or are insufficiently demonstrated.*

**We have updated the manuscript to incorporate also your specific suggestions. In the following you can find our replies point by point to your questions and suggestions:**

**Specific**

*It would be interesting to present the reader with generalities such as: How much does Hamburg represent in national emissions? How is Hamburg ranking in population in Germany?*

**Thank you for this suggestion, we have added information on the population rank to the abstract. Also, we have added a further sentence to the introduction in Line 68 reflecting the relation to national emissions "According to the TNO GHGco inventory 3% of total CH4 emissions in Germany occur in Hamburg (Super et al., 2020)"**

*L17-19 Certainly, I agree that mobile measurements generally miss spatially large, diffuse sources. However, the statement about mobile measurements is inaccurate since 1) comparative capacity of mobile measurements strongly depends on the structure of emissions, and 2) it makes a comparison implicitly with the present study and the present paper did not make a case either that it captured extensively all emissions, or gained temporal representativity.*

**We believe that the last sentence in the abstract is a valid summary of the following paragraph Line 391ff: ". If we consider only fugitive emissions according to the TNO split (A1), our study estimates 210 ±110 kg h−1 for the northern part of Hamburg. This is between two to eight times higher than the estimate presented by Maazallahi et al. (2020). One potential source which is usually not measurable on the street-level, and could thus explain the lower emissions measured by Maazallahi et al. (2020), is end use inside homes (cook stoves, boilers for heating, etc.) (Lebel et al., 2022; Defratyka et al., 2021). Accumulated emissions from end use, while not affecting street-level concentrations, could be observable in total-column measurements and thus contribute
to the higher emission estimates of this study. Another source in Hamburg which could potentially contribute to higher column measurement based estimates is the**

Alster lake near the city centre. Around this lake Maazallahi et al. (2020) have detected CH4 enhancements, which were low in magnitude but spread over a large area. These low enhancements could not be used for quantification and are thus not included in their estimate, but might be noticeable in the column measurements."

In fact, the first part of the abstract sentence ("This study reveals that mobile measurements at street-level may miss the majority of total methane emissions") is a factual statement of the observation in this study. The phrasing "may miss" was used since this study cannot speak for all other cities even though the principle applies elsewhere. The remainder of the abstract sentence describes the likely reasons for this observation. One likely reason is off-street-level emission sources such as leaks from gas end use in homes as described in section 3.4. The other likely reason is that the low signal-to-noise from the Alster lake area from the mobile measurements was insufficient for flux quantification and was in fact not quantified.

In order to make this clearer to readers of the abstract, the following clarifying edits were made in Line 18ff:

"This study reveals that mobile measurements at street-level may miss the majority of total methane emissions, potentially due to sources within buildings including stoves and boilers operating on natural gas. Similarly, the CH4 enhancements recorded during the mobile survey from large area sources, like for instance the Alster lake, were too small to generate GPM emission estimates with confidence, but could nevertheless influence the total column emission estimates."

*L73: it would be more rigorous to oppose biogenic to thermogenic, not "fossil" (fossil CH4 may sometimes have mixed or biogenic origin.*

We have changed that throughout the manuscript according to your suggestion.

*L76 these papers were certainly not the first to "demonstrate" this, however they did successfully use isotopic composition. Please attribute more specifically the usage.*

**Thank you, we have changed the wording in Line 76f to " [...] use the isotopic signature to reveal the source type."**

*L100-103: the inversion framework presented in this paper does not seem to rely on differential columns. This part could be removed or better aligned with the rest of the paper.*

*Strictly speaking the inversion framework would also work with one station measuring. However, we think that to constrain a meaningful background at least two stations should be measuring. This is also why we have a criterion to select days, where at least two stations were measuring at the same time for a certain minimum time.*

**Additionally, we have clarified this in the methodology in Line 101f as follows: "The setup in Hamburg consists of four spectrometers to ensure that for most wind directions the differential column condition is met, and a meaningful background can be constrained by the inversion framework."**

*L112-113: Please provide statistics for differences between instruments during these intercalibrations. Since the inversion relies on small gradients of a few ppb, it would be crucial to be certain that the bias between instruments is well below this order of magnitude. How were spectra processed (at least a reference in section 2.1. would be useful) ? How were the instrument calibrated? Do the instruments benefit from recent advances in COCCON network calibration procedures (Alberti et al., AMT 2022)?*

**We have added Figure A2 in the Appendix showing the four instruments measuring at the same location (side by side) in comparison to a measurement day in network configuration.**

**The mismatch between instruments on a side-by-side day is 0.21+/-0.48ppb.**

**Also we have updated the methodology on Line 126, to give information on the algorithm used in our study.**

**To convert the acquired interferograms into concentrations, we used the GGG2014 retrieval algorithm (Wunch et al., 2015). This algorithm was originally developed for retrieving TCCON data. To adapt it to our EM27/SUN use case, we applied slightly modified spectral windows as described in the EGI setup (Hedelius et al., 2016). The remaining parameters correspond to the TCCON parameters.**

In this measurement campaign, we have not yet applied the recent advances in EM27/SUN calibrations as described in (Alberti et al., AMT, 2022).

*L119 Do different instruments show different number of good days?*

Our definition of good days takes into account all instruments because we would like to have a minimum of two stations measuring to constrain the background and the emissions better. Or in other words we did not calculate the good days for each instrument individually but only for the network.

*L127: "data statistics to remove outliers": how do you define outliers?*

We have added the following sentences for clarification on Line 132:

"In this step measurements are split when for more than 18s no measurement is available. Then each 2 minutes of data are only considered when there is at least continuous measurement data for more than one minute. This way outliers from partial cloud coverage during the interferometer scan are reduced."

*L128: Are gaps >10 min filled?*

We have added the following sentence to clarify this in the methodology on Line 134: "Finally, the remaining continuous measurement sections are averaged using a 10 minutes moving average filter. Gaps are not filled."

*L131: how much data is "lost" in this selection process? Does that lead to the 9 days remaining selected in the end? I would suggest to move the sentence from L119 here.*

During filtering of outliers and the averaging the temporal resolution decreases but the timeline is preserved in general and hence little data is "lost". The selection process of good days has stricter criterions and data here more data is ignored. As the August 2021 was particularly cloudy, several days did not match this criterion.

We have moved the sentence as suggested.

*L149: how do you define the background?*

We have added this information in Line 159f: "The background signal was determined as the lowest 5th percentile of a +-2.5 minutes time window around each data point."

*L155-158: "value": are "values" in mixing ratios or fluxes units? How do you transit from one to the other? How many is "few" measurement points? What do you mean by "suggested"*

**We have updated the manuscript in Line 165ff to clarify this section as follows: "We assumed that in regions where we measured high concentrations, it is more likely to find emission sources than in regions where we measured only background concentrations.**

**The emissions of all inventory pixels that were covered by our mobile survey were summed up and distributed according to the measured concentrations, weighted according to the number of measurements per pixel.**

**In pixels with few measurement points, the new emission value of the pixel was chosen closer to the original value of the inventory. In pixels with many measurement points, the value was chosen closer to the value suggested by the concentration-based redistribution of emissions."**

*L168: please provide here the value of flux estimate from Matousu et al.*

**We have added the value as follows in Line 191: "Matousu et al. 2017 estimated the CH4 flux from the Elbe into the air for different sections of the river between 0.25 and 4.5 kg h-1 km-2"**

*L169: how do you define "plume" (some threshold?)*

**Yes we used a threshold of 100ppb.**

**We have added this information in Line 197: "Plumes were manually selected when an enhancement higher than 100 ppb was observed."**

*L172-177: this appears to be a circular reasoning: adjusting the position according to the assumed flux and then calculating the flux according to the position. Could you please explain how this converges toward a best "position-flux" estimate and how are the uncertainties on the two terms linked?*

**For each location the possible Pasquille-Gifford stability classes are estimated using wind sensor data, information about the daytime and surroundings. In a next step a theoretical $\sigma_y$ of the GPM is determined from the spatial width of the plume of each transect. Under the assumption of a constant stability class, $\sigma_y$ can be expressed as a function of the distance to the source x. This function is**

independent of Q, the flux and thus an estimate for x can be determined.  The distance in combination with the wind direction lead to an estimate of the location for each transect. For each estimated source location, the flux is then estimated. Errors of the location estimates are propagated into the emission estimate.

We have clarified this as follows in the manuscript on Line 200ff: "For this purpose first the possible Pasquille-Gifford stability classes are estimated using wind sensor data, information about the daytime and surroundings. In a next step the width of the plume was used to derive σ_y. Under the assumption of a constant stability class, σ_y can be expressed as a function of the distance to the source. This function is independent of the flux and thus an estimate for distance can be determined. The distance in combination with the wind direction lead to an estimate of the location for each transect. For each estimated source location, the flux is then estimated. Errors of the location estimates are propagated into the mean emission estimate. For each location all relevant Pasquill stability classes were estimated. The presented mean emission estimates are the average of estimates obtained for each relevant stability class and location estimate."

L180-181: I assume this wind measurement is for GPM. Using the wind from elsewhere would generate a very large uncertainty due to wind spatial variability in a city. When available, how do the wind from the mobile sensor compare to the mast?

We have added the following sentence to show how the local wind sensor information compares to the Weathermast on Line 301:
"A comparison of wind data from the local sensor (at 2m altitude) used for the GPM emission estimates and the Weathermast (at 10m altitude) showed a mean difference of 1.6m/s (standard deviation of the difference was 1.2m/s) for the wind speed and a mean difference of 15ºCW (standard deviation 31 ºCW) for the wind direction."

While checking our data again we found that even for the boat measurements local wind sensor data was used. We initially had one more location in the paper for which we used the Weathermast data. This location was however removed in the process. So in fact all presented estimates now use local wind sensor data.

*L195 I would oppose thermogenic and biogenic. Anthropogenic can be biogenic…*

We have updated the sentence to "To analyse the source mix of the measured CH4 and to decide whether it is mainly of thermogenic or biogenic origin, [...]"

*L217: how do you release particles? At evenly distributed altitudes? Up to the tropopause? How do you define receptor station?*

**We have clarified this as follows in Line 242: "Analogous to (Jones et.al 2021, supplement S1) virtual particles are released along the line of sight according to the given solar azimuth and elevation angle in 13 altitudes up to 2220m height above the instrument. These particles are released at the receptor time and travel backwards in time until they reach the simulation border (background time)."**

*L220 I assume following Jones et al., 2021 that footprints are defined up to the BLH and use ERA5. Inverse result would be sensitive to BLH because a lower BLH means that more particles will leave the domain without interfering with local surface emissions. Does the model BLH compare well with wind lidar profiles?*

**We have added Figure A3 in the Appendix, which shows a good agreement between lidar based estimate of the BLH and the ERA5 model BLH.**

*L224 is there time variability allowed for the boundary?*

**The boundary is allowed to vary the background concentration with time. We have updated the sentence as follows on Line 255: "Further assumptions for the model are a spatially homogeneous concentration at the boundary (concentrations can vary with time) and a known spatial distribution of the diffuse emission sources provided by the inventory."**

*L235 "negative days" : please clarify*

**We changed the word "days" to "emissions".**

*L246 : Do you assume only sensitivity to wind direction, and no sensitivity to wind speed error?*

**We have clarified this on Line 280f as follows "No variations were made for the windspeed, as the mean mismatch between lidar and model was as low as 0,49m/s."**

*L252: please clarify how other areas were estimated.*

**We have added the flux as follows in Line 287: "[...] and other areas were estimated with a mean flux of 2.5 kg h-1 km-2."**

*L252: please clarify how is the 8ppb background uncertainty decided.*

**We have clarified this in the manuscript in Line 287ff as follows: "The uncertainty of the background was chosen as 8ppb according to a comparison of MUCCNet measurements with the Copernicus Atmosphere Monitoring Service (CAMS) data, slightly below the value of 10ppb used by Jones et al."**

*L274-279: This may also suggest that the background uncertainty chosen for this study is too optimistic. (as is noted in the discussion, it is difficult to reach a conclusion on this particular day, especially when the authors may be tempted to promote the overlooked river flux as a preferred explanation.)*

**We have also tried higher background uncertainties as visible in the sensitivity study (for the whole campaign period) in Figure A7 in the Appendix. On the 31st of August we got negative emissions with the original inventory even with higher background uncertainties.**

**We think however that a source outside of the domain could also cause such behaviour. Thus, we have added this to the discussion on Line 492 as follows "Alternative reasons for the observed behaviour could also be a too low prior uncertainty or sources outside of the domain."**

*L294: "was weaker": please provide R^2 values here in the text.*

*As Jones et al. (2021) in their final published version do not state a R2 value, we have removed this comparison.*

*Fig 5 Caption and everywhere in the paper: it is misleading to convert fluxes to Gg/y. I would argue that this choice of unit implies in the mind of the reader that upscaling to the yearly fluxes has been done properly, which is not the case, based on 9 days of late summer data. It is necessary to convert to units workable from the data and commensurate to the experiment, e.g. kg/h, mol/s or similar.*

**We have updated the units to kg/h.**

*Fig 5 caption, last line, I suggest to add "prior" before inventory.*

*We have updated Figure 5 according to your suggestion.*

*L303-304: This is very challenging to understand and this need to be explained to the reader. Maybe rephrasing, or a figure would help?*

*We have rephrased that and refered to Figure 3 to explain the example. Please see Line 355ff:*

**"The difference in emission estimates for the three inventory versions on single days, can be explained by the different spatial distributions of prior emissions. In the area covered by the footprint on a particular day, the recorded emissions can be different in the original and the updated inventories. These differences in prior emissions for each inventory version lead to different scaling factors with the same observations. The scaling factor is determined by the inversion when scaling the three inventory versions to match the forward model and the observations.**

**As all emission inventories are normalized and have the same total emissions a different scaling factor, applied to the whole inventory, can lead to different total emission estimates.**

**For instance, the inversion result can differ with original and the modified inventories, when there is footprint covering the industrial zone of the inventory. This zone has higher emissions in the original inventory than in the two updated versions as visible in the lower left panel of each inventory in Figure 3. With the same observations, the scaling factor calculated by the inversion framework will be slightly lower with the original inventory (as the inventory has already higher emissions recorded here), and higher with the updated inventories, as the updated inventories have lower emissions recorded here."**

*L311: section title, I suggest to add "compared to car" or something similar since this is really an interesting dimension of this section.*

**We have updated the section title to "Emission rate estimates from column measurements and comparison to car-based study"**

*L312. How are posterior uncertainties derived? (in other words, how is uncertainty reduction calculated?) Are error reduction maps available to illustrate the weight of network sensitivity?*

**Posterior uncertainties are calculated according to Equation 14 in Jones et al. 2021, based on diagonals of the posterior covariance matrix. As there is no spatial dependency of the error, we cannot produce error reduction maps.**

*L316: this is interesting, when adding the data from 'slowly varying' car-based CH4 measurements, the uncertainty is increasing compared to the 'peak' approach. Any idea why this is the case?*

**Thank you for this good question. One possible reason for this increased uncertainty is the different transport error for each inventory version. The inventory with the peaks has for instance the highest average transport error.**

*L338: Defratyka et al. (2021) did measure furnace/heating CH4 emissions in urban context. The statement after this sentence should be adjusted to reflect this precedent, as well as the fact that small, diffuse emissions will be diluted within the entire column signal in total column measurements.*

**Thank you for pointing this out. We have added the precedent and adjusted the sentence as follows in Line 393ff: "One potential source which is usually not measurable on the street-level, and could thus explain the lower emissions measured by Maazallahi, 2020, is end use inside homes (cook stoves, boilers for heating, etc.) (Lebel et al. (2022), Defratyka et al. (2021). Accumulated emissions from end use, while not affecting street-level concentrations, could be observable in total-column measurements and thus contribute to the higher emission estimates of this study."**

*L351: how many transects were performed? Over how many days?*

**We have added this information to Table 3. Every location was quantified on one single day.**

*L361: "include… in error bar": please be more specific if the statement means that the estimates are not statistically different from each other within their respective uncertainty (and then which uncertainty).*

**We have adjusted the sentence as follows in Line 418ff: "The GPM estimate is not significantly different from the values of the original and the two updated inventory versions (5.4 kg h−1, 15 kg h−1 and 19 kg h−1, respectively)."**

*L375. This quantification was done over how many transects? Over how many days? Generally across the section this type of precision is useful to weigh how the data can be generalized.*

**We have expanded Table 3 to include information on the number of transects. Also we have added the following sentence "All transects for each location were driven on the same day."**

*Table 2. Reading the text (L363-364) indicates that "Type" for L=2 should be undetermined rather than "wwp". This table needs more works (and less abbreviations!). Useful additional columns would be isotopic signature and distance to source, as well as number of point sources.*

**We have added further columns, the distance as well as the number of transects to Table 3. Also we have changed wwp to undefined.**

*L392: estimated by whom?*

**We have clarified this as follows in Line 454f: "The emissions from anthropogenic activity in the administrative region of Hamburg were estimated by this study at [...]"**

*L393: central value for this study's estimate is 30% lower that the TNO inventory, but the 50% uncertainty does not bring sufficient constraint to infirm the inventory. This should be mentioned and pondered in the conclusions of the paper.*

**We have added the following to the conclusion in Line 569ff: "The emission estimate derived in this study has a large uncertainty and estimates from the bottom-up inventories TNO GHGco and EDGAR are not significantly different. Further good measurement days distributed around a year would be needed to get a more certain estimate."**

*L395: There seems to be wishful thinking in the sentence. why "biogenic"? From the inversion it is not possible to say. "potentially the river"… this seems logically desirable but not demonstrated as there are also potentially several other things,*

*including any upstream anthropogenic or natural biogenic source, within or outside the domain that are compatible with the data. The demonstration to single out the 'river' hypothesis is not sufficient. This should be reflected in the formulation.*

*Overall the inversion result is compatible with both EDGAR and TNO inventories, whether or not an additional source (hypothetically the river) is added. This section should be removed or maybe -with the addition of more specific data- kept for the discussion section.*

**We have reformulated the sentence as follows in Line 456ff: "During our study, we observed influence from a biogenic source, which we modelled as river emissions. Large natural area sources such as waterbodies were previously not recorded in the TNO GHGco inventory."**

**Also we have added more specific data to strengthen the river hypothesis in Figure 11.**

*L403. Night peaks and missed peaks suggests that the inversion should use also the in situ measurements since the fair-weather bias of the total column measurements lead to a blindness to high atmospheric CH4 concentrations that may bring complementary information on the fluxes.*

**We have added the following section to the discussion on Line 508ff: "The inversion framework should in the future be developed further to include in-situ CH4 emission data along with column concentrations. This way the modeling could be improved and the inversion could further constrain the emission estimates as well as give more insights if the river emissions could in fact explain the observed enhancements."**

*Fig 9 . panels needs spatial scale for reference. Inventory grid boundaries should also be indicated (the underlying question is: are colors similar between any two adjacent places because they are in the same grid cell or because we have adjacent grid cells with similar values?).*

**We have added the boundaries of the grid cells to the panels as suggested and specified the pixel size in the description as follows: "Inventory pixels are separated by a white dotted line and at this latitude have an approximate length of 1100m and a width of of 650m." As some of the panels are in perspective this way**

**the spatial scale is given for objects in the foreground and the background.**

*Does Fig 12 show all data over the same period as Fig 11?*

**Both Figures refer to the same period.**

*Section 3.7 The isotopic signature is driven by peaks in Fig 11. The peaks (difficult to see on the small figure) might be of very local origin and therefore the isotopic signature would not be of significance at the urban scale. If it were significant, then it is problematic that the EM27 were not measuring when these peaks are observed. L412-413. Could you please expand on other potential sources contributing to the dD signature? This would be useful to clarify whether the river hypothesis can be sustained through the conclusions of the paper.*

**We have added a sentence on the canals in Hamburg to the discussion in Line 475ff: "The sharp short-term peaks could be caused by canals in the city close to the in-situ instrument, that fall dry during low tide and then fill up again during high tide."**

*Section 3.8 I suggest to move this section before the inversion results because this is supporting the confidence in the inversion.*

**We moved this section.**

*L419 Is there any difference expected between the wind that surface emissions are seeing and 100m+ wind? Is there any chance to use lower 2m or 10m wind measurement for validation?*

**Probably there are differences between surface winds and wind at 100m especially because of turbulences around buildings ect. However at a modeling resolution of around 1.1km by 0.6km these should average out on grid cells and surface wind should in general follow the direction at 100m.**

**A comparison with data from a single sensor on the weather mast (co-located with the lidar) is possible. However the representativeness of such a comparison for the whole city (in different locations of the city turbulent flow at 2m and 10m depends on the buildings near by) is not given.**

*L430 if there is a source that cannot be updated in the prior, it can have alternative reasons: 1) too low prior uncertainty (preventing the inversion to push upward) or 2) source outside of domain. In any case if the cost function attributes the difference to the background it shouldn't be a problem. It makes no sense that it pushes the background too high and at the same time pushes the within-domain fluxes to negative values. It looks more like a poor parametrization of the cost function or a suggestion of improvement of the code.*

**Thank you for your comments and pointing out further reasons that could lead to the described behaviour.**

**We have added these to the discussion as follows in Line 492f: "Alternative reasons for the observed behaviour could also be a too low prior uncertainty or sources outside of the domain."**

*L475 the wide error bars in Fig 10 tend to suggest otherwise: the method leads to uncertainties that are much larger than the uncertainty required to verify or improve the inventory. Moreover, on a separate note, the strong sensitivity of the small-domain inversion system to prior fluxes and potential boundary effects limits the confidence in the inversion system. The presentation of the conclusions should reflect this.*

*We have changed the presentation in the conclusion to reflect that as follows in Line 569ff: "The emission estimate derived in this study has a large uncertainty and estimates from the bottom-up inventories TNOGHGco and EDGAR are not significantly different. Further good measurement days distributed around a year would be needed to get a more certain estimate. Also further improvements to the small-domain inversion system could be made to exclude the possibility of the boundary conditions affecting the emission estimates."*

*L521 again some authors did find a non-negligible contribution from boiler rooms based on mobile measurements.*

**We have added a reference here as well.**

**Editorial**

*Please double check all reference formatting in the text according to journal rules.*

**We have checked the references in the text and updated the formatting according to journal rules.**

*Please avoid unnecessary abbreviations in figure labels; e.g. in Fig 5 replace "upd:elv" by "Updated, peaks",*

**We have chosen to use this abbreviation because of Figure A6 in the appendix. Here longer labels would be difficult to integrate in the figure. To be consistent throughout the paper and make it easier for the reader to understand what inventory is meant, we have used the same abbreviation elsewhere.**

*in Fig 6 replace "X1:Anthrop." By "Anthropogenic".*

**We have updated this**

*If Fig. 4 remove "em" ("em" is not even explained in the caption). Readable words are mindful of the reader and would avoid multiple searches in figure descriptions and in text.*

**We removed "em".**

*Similarily, in tables there is ample room to use full words instead of unnecessary abbreviations. Similar examples as for figs, also as example expand fully "W Spd Model Mism", this will still fit in the page. Etc.*

**We have updated the table with whole words.**

*Start axes labels with capital letters (see e.g. Fig 10)*

**Labels were capitalized.**

*L16 Please harmonize significant digits (e.g. 14.0+/-8.0)*

**We have harmonized all emission estimates to two significant digits.**

**Isotopic signatures are shown with three significant digits as in comparable literature.**

*L24 Reference could be updated*

**We updated the reference.**

*L27-28: "for instance a typical...": very vague wording, please be specific or simplify the sentence.*

**We have updated this sentence**

*L32-33: no capital letter to \*M\*ethane*

**We have corrected that.**

*L41: remove "fossil... to as" (unnecessary and not precise)*

**We have corrected that.**

*L48: unclear sentence. Short solution, insert "this" in "and \*this\* gradually lead...". Better solution, reconsider the full sentence for sharpness.*

**We have updated that.**

*L53: replace fossil gas by natural gas*

**We have updated that.**

*L67: suggest to reference document as Hamburg port, 2021.*

**We have updated that.**

*Fig 1. What is the meaning of the grey shadings? Isotope triangle not visible on figure. Please adjust.*

**We have adjusted the description and the triangle.**

*L92-98: this paragraph seems to be somewhat redundant with information given in the introduction, and could be shortened.*

**We have shortened this paragraph.**

*L183: I suggest to refer to Fig 1 here.*

**Reference added.**

*L261: what should be compared to Fig 5? Ambiguous sentence.*

**We clarified that.**

*Fig 4: Difficult to read. I suggest to limit the x axis after 6am. It will help readability of the figure by expanding the x dimension, and in some cases also the y dimension.*

**We updated Figure 4 according to your suggestions.**

*Fig 6: panels a and d are poorly readable (see comment just above). Also, b and e appear to be 99% redundant and I can't see the river dashed delineation unless I zoom. I suggest to remove b and e.*

**We updated Figure 6 according to your suggestions. We would like to keep panel b and e though, because one can see the footprints, which help the reader to understand why on this day mc could have been sensitive to river emissions.**

*L367. reference to Fig 9 applies to other paragraphs above this one. Please move upward this reference.*

**We moved the reference.**

*L388-390. This seems to wrap up the finishing section, this should be made clearer (or simplified)*

**We have added some clarification and quantitative detail here in Line 446ff: "In general several significant CH4 sources were quantified during the mobile survey. While several GPM estimates confirmed the values recorded in the emission inventory (both updated and original versions), some of the biogenic and thermogenic sources estimated using GPM, like Location 1 and 5, were significantly above the values recorded in the TNO GHGco inventory.**

**The correlation between GPM estimates and the inventory values is highest for the Upd:elv inventory, $R^2$ = 0.13 compared to $R^2$ = 0.10 and $R^2$ = 0.10 for the Upd:all and the original inventory, respectively. On the other hand side, the Root Mean Square Error (RMSE) is highest for the Upd:elv inventory, 27 kg h-1 compared to the Upd:all and the original inventory with a RMSE of 4.4 and 4.1 kg h-1, respectively."**

*L392 and Fig 10: are "administrative region of Hamburg" (L392) and "City of Hamburg" (Fig 10 caption) identical entities? If yes please harmonize the names, and if no please explain the difference.*

**We have harmonized the names.**

*Fig 9. There is a lack of caption title.*

**A title was added.**

*Fig 10. Light colors against light grey background are not visible. Please change colors to remain readable for a majority of readers (and I strongly suggest to use a clean white background in figures to reduce the feeling of visual clutter.)*

**We have changed the background to white for Figure 10.**

*I suggest to merge fig 11and 12.*

**We have changed Figure 11 and 12 to show more detail. Merging would make the figure too large for one page.**

*L419. Please clarify until (above? Below?)*

**We have removed this sentence because it does not add any meaningful details.**

*L424 Please clarify what is a "standard" inventory*

**We have added "standard inventories, such as the TNO GHGco inventory"**

*Fig A4: remove "this plot shows the" from the caption*

**We have removed this.**

*Fig A5: This figure is a complex representation and should therefore be thoroughly explained. What are the radial distance numbers? What is the unit? does the figure correspond to a particular level or levels- and which ones?*

**We have updated the description accordingly.**

*Appendix Tables: please provide complete table captions, remove text on the side, and use full words instead of abbreviations.*

**We have changed the tables according to your suggestion.**

**Reply to Referee Comment 2**

*The manuscript presents an interesting case study on the use of an array of solar spectrometers and mobile methane measurements to improve the methane emission estimate of a city, Hamburg in this case. The paper is mostly well written and logical in structure. Most of the conclusions are soundly based on the evidence. However, there are a few shortcomings in the manuscript in its current form. First, the methodology is at some parts too vaguely described whereas icn other parts commonly known methods are described in unnecessary detail. All of the conclusions a not very convincing based on the evidence shown. Therefore I suggest major revisions before reconsidering the manuscript for final publication in the ACP. Please find my detailed comments below.*

**Thank you for your thorough analysis of our manuscript and your suggestions. We have incorporated your feedback into the manuscript and reply in line with your comments. Please find our replies are in bold.**

Detailed comments

*Page 1, line 10: "...mobile measurements did not have a significant effect on the total emission estimates for the campaign period."*

*Page 1, lines 12-13: "...highlighting the potential of mobile measurements to derive up-to-date emission inventories.*

*These two sentences in the Abstract seem to be somewhat contradictory. Maybe a clarification is needed.*

**We have clarified this as follows in Line 11ff:"A comparison of the updated inventories with emission estimates from a Gaussian Plume Model (GPM) showed that the updated versions of the inventory in several cases match the GPM emissions estimates well, revealing the potential of mobile measurements to update the spatial distribution of emission inventories."**

*What is the justification of the location of the spectrometers? It seems that the spectrometer in the north is not at very optimal location as there are a lot of a priori sources north of it, according to Figure 1.*

**We have clarified our reasoning as follows in Line 122ff: "The northern site was co-located with the isotope measurements on top of the roof of the Geomatikum building of the University of Hamburg. This location was chosen weighting different criterions firstly the availability of sites with suitable conditions to house the room-size setup for isotopic measurements as well as allowing for the set up of the FTIR instruments on top of a flat roof. The Second consideration was to place the site outside of the industrial area, a high emission zone according to the TNO GHGco inventory."**

*Page 5, lines 109-111: "For this study yearly average emission estimates as recorded in the inventory were considered. Between the 27th of July 2021 and the 9th of Sept 2021 our four FTIR-Spectrometers were measuring in Hamburg. From 30 July to 5 Sept, the instruments were deployed at different locations." Is this a source of bias, as the inventory is annual averages but the measurement campaign is in summer (with less heating etc). This should be at least discussed more extensively.*

**We have added the following sentence to the discussion in Line 551ff: "Also the prior emission inventory is based on average yearly emissions (summer and winter months), thus the prior emissions could not be fully representative of the study period in the summer."**

**Also we have changed the units to kg h-1.**

*Page 6, lines 152-154: "...both the complete signal (background and enhancement peaks, later referenced as "upd:all") as well as the peaks only (later referenced as "upd:elv") were averaged on the inventory grid, as can be seen in the right and central plot of Figure 3." It is not clear from this how the emission is derived from concentration data, and how exactly "upd:all" and "upd:elv" are derived differently. It seems the general description of derivation is in page 7, lines 169-176. That should be moved right after this sentence, with addition of how exactly "upd:all" and "upd:elv" differ.*

**Thank you for letting us know that this is still unclear.**

**We have added the following sentences for clarification in Line 165ff: "We assumed that in regions where we measured high concentrations, it is more likely to find emission sources than in regions where we measured only background concentrations.**

**The emissions of all inventory pixels that were covered by our mobile survey were summed up and distributed according to the measured concentrations, weighted according to the number of measurements per pixel."**

*Page 6, lines 157-158: "In pixels with few measurement points, the new value of the pixel was chosen close to the original value of the inventory. In pixels with many measurement points, the value was chosen closer to the value suggested by the mobile measurements." Can you be more specific on how this is done, e.g. by providing equation?*

**Thank you for your suggestion, we have now added the relevant equations to the manuscript. Please see Equations 1-4.**

*Page, 6, lines 159-160: "For a better comparability between the updated and the original inventory, the sum of emissions in the area covered by our mobile measurements is equal in the original and the updated versions." Is this achieved by normalization? Can you provide details on how this is done, e.g. equation?*

**As for the question above we have clarified this with Equations 1-4.**

*Page 8, lines 185-186: "For each measurement day a standard deviation of the wind direction and speed was derived." Is this the standard deviation of measured wind or also differences between measured and ERA5 winds?*

**We have clarified this as follows "For each measurement day the standard deviation of the differences between ERA5 model and the lidar wind direction and speed was derived." in Line 215f.**

*Page 9, lines 216-217: "It results in a nearly Gaussian shaped distribution of background time for each receptor time." What is this Gaussian distribution used for and how?*

**The number of particles passing the boundary border has a Gaussian distribution, i.e. there is a background time most particles pass the border. Before and after that time, there are less particles passing the border.**

*Page 9, line 218: "Releasing particles backwards in time is also the basis to generate footprint matrices..." How exactly were the footprint matrices generated?*

**We have added the following clarification in Line 249f: "The footprint is the summation of the residence time of all the particles in a grid cell."**

*Page 10, line 244 – page 11, line 253: It is unclear how the sigma values relate to the equation (4).*

**We clarified this in Line 289f as follows: "Sigma_sector^prior and sigma_background^prior are the diagonal elements of S_a as in Jones et al 2021."**

*Page 11, lines 270-280: So, if the prior is very wrong the column measurements may result in artificial results, even negative emissions. Did you do any sensitivity analysis by e.g. varying the weights given to prior and observations in Equation (3)?*

**We have done a sensitivity study of several inversion parameters and have now added a plot in the Appendix Figure A7.**

*Page 12 line 284: "...decided to include the Elbe river in all other model runs and days presented in this paper." Is it included in Figure 5?*

**Yes it is included for all inventory versions of Figure 5 as stated in the caption "The emission of the river Elbe was added to all versions of the emission inventory"**

*Section 3.5: I find this whole section very confusing. Only values of emission derived by GPM (?) are given, not the values they are compared to. The uncertainties are very large, exceeding 100% at some cases. Please provide more quantitative detail to comparisons.*

**We have added more quantitative comparisons here.**

*Also, the Figure 9, associated with this analysis is very pretty, but very confusing and not quantitatively informative.*

**Figure 9 gives qualitative information on the site locations and the surroundings for the reader to better understand the study sites for quantitative information please refer to Table 3.**

*Page 18, lines 392-393: "The emissions from anthropogenic activity in the administrative region of Hamburg was estimated at 7.9 ±4.4 Gg/yr which is lower than the 12 Gg/yr reported in the TNO GHGco inventory for the year 2015. This difference is however not significant and the inventory value falls within the uncertainty of this study." You can't state one value to be lower than another if the difference is not significant. I suggest rewording: "The emissions from anthropogenic activity in the administrative region of Hamburg was estimated at 7.9 ±4.4 Gg/yr which is not significantly different from the 12 Gg/yr reported in the TNO GHGco inventory for the year 2015."*

**Thank you for your suggestion we have adopted your propsal.**

*Page 18, line 395: "During our study, influence from a biogenic source, potentially the river, was found." I could argue that the influence of the river was not found in this study, but corroborated.*

**We have added a further plot (please see Figure 11) to show the correlation between the measured CH4 peaks and the rising tide of the river estuary. This correlation supports the river hypothesis.**

*Page 19, line 404-405: Could the peaks and isotope signals during different nights and possibly wind directions be analyzed separately to detect possible differences in source signature and source apportionment?*

**We have added Figure 11 to show how the insitu CH4 measurements correlate with the waterlevel in the river. We have also added information on the wind direction in this plot.**

*Page 19, line 409 – page 20, line 410: "...less enriched..." Would "...more depleted..." be better expression as the delta values are negative?*

**Thank you we have changed that according to your suggestion.**

*Page 23, line 475-476: "...turns out to be a promising technique to update city scale emission inventories." Is it honestly so? I found the fact that the emissions shown in Figure 5 are all over the place not very convincing, along with the large uncertainties. You should justify this statement better.*

**We have updated the wording in Line 537 to "Scaling this updated map according to the findings of an inversion framework (using column concentration measurements), turns out to be a feasible technique to update city scale emission inventories. To yield representative emission inventories this approach would need to be carried out for a longer time period, than in the present study, though."**

*Page 24, lines 499-500: "This study shows that the CH4 emissions of a large source region like the municipal area of Hamburg can be quantified with FTIR Spectrometers and inverse modeling." As the measurement data with inversions did not change much the prior emission, I find this statement not really supported by the results. If you would have arrived to a reasonable emission by this measurement and inversions alone, I could agree. Now it seems that the prior was controlling the end result.*

**We have updated the sentence as follows in Line 564: "This study shows the challenges of quantifying CH4 emissions of a large source region like the municipal area of Hamburg."**

Technical:

*In many places the numerical values are presented with unnecessary precision (i.e. with too many significant digits). A good guide to the usage of significant digits can be found in e.g. R.J. Taylor, An Introduction to Error Analysis, University Science Books, 1982, Chapter 2. A couple of examples on this follows:*

*Page 1, line 14: 0.069 ± 0.047 should be 0.07 ± 0.04.*

*Page 1, lines 16-17: 7.9 ± 4.4 should be 8 ± 4.*

**Thank you for your suggestion, we have unified significant digits as follows:**

**For emission estimates we have chosen to use two significant digits according to the EPA standard (https://www.epa.gov/emc/technical-information-document-024-memo-rounding-and-significant-figures)**

**Isotopic signatures are presented with three significant digits, to match the precision given in related literature that we compare our signatures with.**

*Page 2, lines 27-29: "Fossil fuel CH4 emissions are for instance a typical anthropogenic source, such as fugitive emissions from gas pipelines contribute significantly to anthropogenic CH4 emissions (Schwietzke et al. (2014); McKain et al. (2015)), or road transport and combustion of CH4 (Defratyka et al. (2021))." The beginning of the sentence (before first comma) and the end (after the comma) do not fit together.*

**Thank you we have corrected that.**

*Similar broken sentences are also in other locations. E.g. page 2 lines 48-49: "Isolated CH4 sources can be quantified best individually and gradually lead to a better understanding of mix of sources in a certain area." I suggest proofreading the manuscript very carefully.*

**Thank you we have updated this for clarity.**

*Page 2, lines 32-33: "Methane" should not be capitalized.*

**We have corrected that.**

*Page 5, lines 108-109: "TNO GHGco is currently the highest resolved GHG emission inventory that is available for Hamburg." would be better as "TNO GHGco is currently the highest resolution GHG emission inventory that is available for Hamburg."*

**We have updated that according to your suggestion.**

*Page 5, line 111: "...the instruments were deployed at different locations." Would this be better and correct as "...the instruments were deployed at locations shown in Figure 1."*

**We have updated that according to your suggestion.**

*Page 6, lines 143-144: "All tracks can be seen in Figure 2." Figure 2 is so small that one cannot really see the tracks.*

**We have increased the size of the map.**

*Page 9: The equations for delta notation are quite standard, so they can be omitted with just a reference to a suitable source.*

**Thank you we have removed the equations and referenced suitable sources.**

*Page 9, lines 203-204: "The dominant source type that is responsible for the observed CH4 elevations above background in Hamburg was obtained from a Keeling plot analysis (Keeling (1958))." This is bit inexact. The Keeling plot yields the delta value of the source but not the source type. For source type characterization of the measured delta value is cross referenced to known delta values of different emission sources. I suggest reformulation of this sentence.*

**We have reformulated this accordingly on Line 230ff: "The dominant source type that is responsible for the observed CH4 elevations above background in Hamburg was obtained by comparing delta D and delta 13C values obtained from a Keeling plot analysis (Keeling (1958)) to similar sources signatures in the literature."**

*Page 10, line 224: "...spatially homogeneous concentration at the boundary..." Does this refer to the boundary of the modeling domain?*

**Yes, we have clarified this. Please refer to Line 255.**

*Page 12, Figure 4: It would be interesting to have wind directions shown in this figure as well, is possible.*

**We have added the hourly wind directions to Figure 4 as suggested.**

*Page 16, line 351: "These emissions were quantified as 69±47 t/yr..." Why is there a different unit here than everywhere else?*

**We have now unified units throughout the manuscript.**

*Page 18, line 403 – page 19, line 404: "...Keeling plots indicate a biogenic origin..." A more exact sentence would be "...Keeling plots yield source signatures which indicate a biogenic origin..."*

**We have updated this according to your suggestion.**

*Page 20, Figure 10: The lightest colors (especially red) is almost invisible when printed. Maybe removing background grey color could help.*

**The grey background was removed according to your suggestion.**

*Page 23, line 465: "The variability on single days..." Maybe more exact expression could be "The variability between individual days..."?*

**We have updated that. Please refer to Line 528.**